# Illumination of Interior Spaces through Structures Made of Unified Slabs of High-Performance Light-Transmitting Concrete with Embedded Optical Fibers

**DOI:** 10.3390/ma16083142

**Published:** 2023-04-16

**Authors:** Nikola Štochl, Jaroslav Vychytil, Petr Hájek

**Affiliations:** 1Department of Architectural Engineering, Faculty of Civil Engineering, Czech Technical University in Prague, Thákurova 2077/7, 166 29 Prague, Czech Republic; 2Metrostav a.s., Divize 3, Construction Company, Koželužská 2450/4, 180 00 Prague, Czech Republic

**Keywords:** light-transmitting concrete, plastic optical fibers, light-transmitting construction, translucent concrete, light-permeable concrete

## Abstract

Light-transmitting concrete as a building material already exists in many forms, but its light properties and the possibilities of using it to improve the lighting of interior spaces have not been investigated in detail yet. This paper focuses on the illumination of interior spaces using constructions made of light-transmitting concrete, which will allow light to pass between individual spaces. The experimental measurements carried out are divided into two typical situations using reduced room models. The first part of the paper focuses on the illumination of the room through the penetration of daylight through the ceiling made of light-transmitting concrete. The second part of the paper investigates the transmission of artificial light from one room to another through a non-load-bearing dividing structure composed of unified slabs of light-transmitting concrete. For the experiments, several models and samples were created for comparison. The first step of the experiment was to create slabs of light-transmitting concrete. While there are many options to produce such a slab, the best option is to use high-performance concrete with glass-fiber reinforcement, which improves the load transfer properties, and plastic optical fibers for light transmission. By adding optical fibers, we can achieve the transmission of light between any two spaces. For both of the experiments, we used reduced-scale models of rooms. Slabs with dimensions of 250 × 250 × 20 mm and 250 × 250 × 30 mm were used in three versions: concrete slabs with optical fibers, concrete slabs with air holes and solid slabs. The experiment measured and compared the level of illumination at several points in the model as it passed through each of the three different slabs. Based on the results of these experiments, it was concluded that the interior level of illumination of any space can be improved by using light-transmitting concrete, especially those without access to natural light. The experiment also assessed the strength properties of the slabs in relation to their intended use and compares them with the properties of stone slabs used as cladding.

## 1. Introduction

The construction industry today offers a huge range of building materials that can be combined in various ways to create an entirely new material for a specific purpose as the need demands. This is how light-transmitting concrete came to be created [1]. Light is a very important factor that affects a person’s mental and physical health. But today its abundance is diminished in cities where construction is spreading to the borders of already built-up areas. In addition, the buildings are designed to utilize every square meter possible within the boundaries of the building codes. Translucent concrete is already invented and has been researched from many different points of view [2]. For example, Litracon consists of a bundle of optical fibers placed in a block, which is then separated into slabs [3]. Civil engineering companies Lucem and Luccon also manufacture light-permeable concrete, but the fibers are placed more precisely in the matrix [4,5]. On the contrary, the designer company Gravelli, which was created the material LiCrete, uses a plastic matrix, unlike the previously mentioned companies [6].

This paper focuses on the experimental measurement of light transmission through light-permeable concrete elements, the use of which would allow us to increase the amount of visible electromagnetic radiation into the interior of buildings. The main objective of this paper is to measure the improvement of lighting conditions inside a building using light-transmitting concrete structures. The other part of the study is to comprehensively create and assess samples of a material capable of improving the illumination of interior spaces with daylight and at the same time bring daylight into communal spaces (like lobbies, corridors, stairwells) through walls made of light-transmitting concrete. The originality of this study is in the measurement of light conditions in interior spaces and the light properties of slabs of light-transmitting concrete, which are also investigated in terms of strength for their purpose of use. This study also deals with the technical analysis of the material and its production, as, so far, its properties have not been assessed in this way. There are several processes for making light-transmitting concrete [7,8]. The design of slabs made of light-transmitting concrete is a very difficult and complex task, which includes the analysis of many parameters [9]. These include the ultimate bearing capacity, the ratios of the components of the concrete mixture, the type of reinforcement for the chosen purpose of use, the composition of the mixture in relation to the use of optical fibers and the prevention of their damage. The strength of the concrete mixture itself is already well known, but for our measurements, the commonly used mixture in construction had to be modified due to the addition of optical fibers and air holes. The addition of optical fibers to the concrete mixture does not significantly affect its final strength [10]. The composition of the concrete mixture was influenced, for example, by the aggregate fraction, its chemical composition, its workability, and color. The concrete mixture designed for our purposes was tested by standard procedures and on test samples. The correct composition of the concrete mixture, the volume ratio, the choice of optical fibers and their distribution in the structure significantly influence the measurement results. The ratio of optical fibers to the compound is very important [11]. A larger amount of optical fibers increases the light gains, but at the same time increases the cost of the product itself [12,13]. It may seem that the placement of optical fibers in concrete cannot significantly affect the quality of the indoor spaces from the point of view of illumination, but the measurements performed show that the values are several times higher than when we use a same-size sample from the same concrete mixture, with the optical fibers replaced only by continuous holes of the same dimension. Light radiation moves through the circular air penetrations of the slab at a significantly lower intensity than through optical fibers. Unsurprisingly, when using a solid slab, the light does not spread between the two spaces at all. Experimental test measurements were performed on several concrete slabs from the same concrete mixture using the same type of glass-fiber reinforcement. For the purpose of this experiment, three types of concrete slabs were chosen: (a) solid; (b) with the addition of optical fibers of thickness 3 mm in a precise grid; and (c) with air holes of the same diameter and in the same grid as that of the optical fibers. The experimental study had to be divided into two parts. The individual phases define the type of material, such as the composition of the concrete mixture, its reinforcement and the use of the type of optical fibers. Next, the production of unified slabs and the various light parameters used in the measurements are factored into the final conclusions. The research focuses on the possibility of using light-permeable concrete in places where daylighting is not possible using standard structures such as windows, skylights, etc. The standards specify a minimum daylight factor value that we are able to achieve using light-transmitting concrete in locations where this would not otherwise be possible. The objective of this study was to measure the light properties of light-transmitting concrete and evaluate its possible use in improving the lighting conditions inside a building, which are important for human mental health. The other objective of the paper was to measure the strength properties of light-transmitting concrete slabs and compare them with those of well-known stone slabs, such as travertine and granite. We compared these because the slabs could be used in the same way as stone cladding.

## 2. Theory and Definition

### 2.1. Theory of Light-Transmitting Concrete

The first mention of light-transmitting concrete dates back to 1922, when a German scientist, Paul Liese, filed a patent application on the subject in the United States. Mr. Liese was involved in the development of concrete panels and blocks for vertical and horizontal structures. He was finally granted a patent in 1925 [2]. Since that time, numerous studies and scientific researchers have been involved in the development of light-permeable concrete. The first to invent light-transmitting concrete, as we know it today, was the Hungarian architect Aron Lasonczi. In 2001, he used optical fibers randomly mixed into a concrete mixture to create a block of concrete, which was then cut into individual slabs of specified dimensions. Lasonczi’s light-transmitting panels are what we know today under the brand name Litracon [14]. His method is not the only one. In 2009, Mexican scientists Gutiérrez J. S. and Cázares S. O. G. filed for patent approval for translucent concrete that lets through 80% of the light and weighs only 30% of standard concrete [15]. There are several other ways to create light-transmitting concrete; for example, one scientific team made it by randomly placing the optical fibers directly into a long beam-shaped sample and then cutting it into individual slabs; another team produced individual slabs directly, with the placement of fibers in individual precise positions [16,17,18,19]. There are already several companies that are engaged in the production of translucent concrete. In addition to the Hungarian company Litracon, there is, for example, the German company Lucem or the Austrian company Luccon [3,4,5]. One option is adding a light-enabling element throughout the concrete mixture. Usually, this element is placed into a precisely spaced matrix. Elements that can be considered for use include plastic forms, glass elements (which must be alkali-resistant), photosensitive materials or, in the optimal case, optical fibers (glass or plastic) [20]. Research done using plastic pipes or rods concluded that a large amount of money could be saved by using the plastic optical fibers [21,22]. Further research has shown that the amount of light that optical fibers are able to transmit is significantly greater than that of plastic pipes or rods [23]. The above-mentioned elements allow for the transmission of the entire light spectrum, including the transmission of infrared radiation, which provides heat losses and gains [24]. One of the most famous and spectacular structures using light-transmitting concrete is the Italian Pavilion for EXPO 2010 in Shanghai. It was built by the Italcementi Group using a material created known as i.light [25]. I.light is a special concrete mixture to which a plastic matrix has been added, enabling the transmission of light from the exterior to the interior [26,27].

### 2.2. Physical Definition of Solar Radiation

The sun is a source of electromagnetic radiation in a wide range of wavelengths, from gamma rays *λ* = 1 pm, x-rays *λ* = 10 pm–1 nm, ultraviolet *λ* = 10 nm–1 µm, infrared *λ* = 10 µm–1 mm, to radio waves *λ* = 10 m and more meters. However, only part of this radiation reaches the earth’s surface. Most of the solar radiation is deflected by the Earth’s magnetic field or absorbed by the atmosphere. As can be seen in the Figure 1, only radiation with a range of 100 to 1400 nm (so-called optical radiation) falls on the earth’s surface. Optical radiation is usually divided into three parts: ultraviolet radiation (UV); visible radiation (VR); and infrared radiation (IR). An air mass (AM) is a coefficient used to define the light spectrum at mid-latitudes through the Earth’s atmosphere. AM1.5, i.e., 1.5 time atmosphere thickness, corresponds to a solar zenith angle of *z* = 48.2°. The AM number is necessary for the overall yearly average for mid-latitudes [28]. It has been determined that AM1.5 global irradiation mainly consists of visible light (53.5%), followed by IR radiation (43.2%) and UV radiation (3.3%) [29]. Ultraviolet radiation has positive effects on human health, as is necessary for the production of vitamin D, but excessive exposure to UV light can cause degenerative changes in the skin or cause the formation of malignant tumors. Within optical radiation, radiation visible to the human eye with wavelengths of 380 to 760 nm is perceived as a spectrum of colors from violet to blue, to green, to yellow, and red [30]. The electromagnetic spectrum and its wavelengths are explained in Figure 1. Visible light is essential for stimulating sensation in the optical nerves, allowing us to perceive the world around us. The last component of optical radiation is infrared radiation. We perceive this radiation on the surface of the body in the form of heat. Regarding building interiors, in the summer, radiation produces unpleasant overheating, but in the winter, it contributes to the heating of buildings. Thermal losses and gains are one of the positive properties of light-transmitting slabs [31]. This has been proven by a study of translucent concrete panels (TC), which shows us that the TC panels can reduce energy expenditure by 18% for a fiber volumetric ratio of 5.6%. This demonstrates the practicality of fabricating TC panels [32]. Due to the high demand for energy in the construction industry, along with its increasing price and market instability, it is crucial to solve the problem of how to reduce these costs. Energy consumption in the industry is estimated at 30–40% of total world consumption [33]. Even after its completion, continued operation of a building is an energy drain, with artificial lighting consuming 19% of the total electricity supplied worldwide [34,35]. Studies claim that buildings covered with translucent concrete panels consume less energy thus lowering costs and speeding up payback time [36].

## 3. Materials and Sample Production

### 3.1. Concrete Mixture

Portland cement, which was used for the production of samples, is one of the most widely used types of cement [37]. The proportions of the ingredients in the mixture can vary depending on the intended use of the concrete, which is why we changed the proportions for each sample. The concrete mixture used in this study is PZ-HPC-53 with a bulk density of 1975 kg/m^3^, without plastic optical fibers. Due to the resulting density of the mixture, in our case it was a lightweight concrete with HPC strength parameters. The exact specification of the used mixture is given in the Table 1 and the mixture itself is shown in the Figure 2. Both the compressive and tensile flexural strength of our mixture was determined by standard concrete tests. The resulting measurements determined a compressive strength of 59.8 MPa and a tensile flexural strength of 10.7 MPa. The size of the aggregate should not be greater than 4 mm [38]. Instead of aggregate, recycled materials such as waste glass from the flat glass industry can be used to improve light transmission, but the strength of the resulting element would be reduced [39]. Because of that, we did not use waste glass for our samples. Glass fibers were added and thoroughly mixed into the concrete mixture, as shown in the Figure 2, and were classified as non-reactive in the accelerated alkali-silica reaction test [40]. For the production of concrete slabs with dimensions of 250 × 250 × 20 mm or 250 × 250 × 30 mm, white Portland cement of class 52.5 R was used. Aggregate passing through a sieve with a diameter of 4 mm was used in the concrete. It was important to choose the right superplasticizer for our mixture in order to have a well-workable mixture. Bright microsilica and a water coefficient of 0.29 were also used with respect to superplasticizer. The samples with optical fibers, which were evaluated in our experiment, had a ratio of optical fibers to concrete mixture of 1:10. The volume of optical fibers in the sample was 10.2% (~120 kg/m^3^) of the total volume. One study investigated the production of translucent panels using a resin-concrete mixture, which could be used as an alternative mixture [41].

### 3.2. Glass Fiber Reinforcement

The use of glass fibers is necessary, because of the purpose of the slabs (façade, cladding) and to increase their strength. The strength of the slabs is slightly reduced, and the material is more brittle because of the raster network of optical fibers or air holes. Anti-Crak^®^ HP (62.4) [42] fibers were used to produce concrete samples with optical fibers. These Anti-Crak^®^ HP (High Performance) glass fibers, the specifications of which are in Table 2, are technologically perfect high-strength, high-modulus chopped strands from Cem-FIL alkali-resistant (AR) glass fibers intended for reinforcing cement-based materials. The strands of Anti-Crak^®^ HP fiber have an optimized special surface treatment guaranteeing the resistance of the strand against abrasion while also maintaining the integrity of the strand during the mixing of the concrete. These integral strands are made up of 100 pieces of individual fibers with a low longitudinal weight equal to 45, and therefore their reinforcing effect is very high even at low weight doses [42]. Reinforcing glass fibers have approximately the same bulk density as concrete and therefore do not sink to the bottom of the mixture nor float to the surface; their shape is shown in the Figure 3. They enable a very homogeneous dispersion of the strands throughout the volume of concrete. This type of fiber was designed to improve the mechanical properties of concrete mixtures, especially its tensile strength, impact strength and ductility. They can replace steel reinforcement nets or wires and prevent cracking. They do not corrode and do not need cover layers. At the same time, they do not negatively affect the workability of the concrete mixture, which is essential to produce slabs with optical fiber distribution in a very dense grid arrangement. It is critical that the material from which the glass fibers are made be alkali-resistant [43]. Safe and easy handling in combination with their excellent mixability is another positive feature of glass fibers.

### 3.3. Optical Fiber Light Transmission

An optical fiber is a cylindrical dielectric waveguide in which electromagnetic waves (usually light or infrared radiation) propagate in the direction of the fiber axis using the principle of total reflection at the interface of two media with a different refractive index [44]. The inner part of the fiber is called the core, and around the core is the sheath, its primary protection. To couple the optical signal to the core, the refractive index of the core must be higher than that of the cladding [45]. Although both glass and plastic are transparent at particular wavelengths, allowing the fiber to guide light efficiently, major differences exist between the two materials when it comes to making the optical fiber. The differences between the POF and GOF fibers can be seen in the Figure 4. Plastic-core fibers are more flexible and inexpensive than glass ones. The bulk density of plastic optical fibers is around 1430 kg/m^3^ [8]. They are easier to install, can withstand greater stresses and weigh 60% less than glass fibers [14]. Plastic Optical Fiber (POF) shown in the Figure 5, is a plastic fiber that transmits signals in the direction of its longitudinal axis by means of visible electromagnetic radiation. Most POFs in use are 1000 µm in diameter, of which the core itself is up to 980 µm. Thanks to this diameter, transmission is possible even if the ends of the fibers are slightly dirty or damaged, or if the axis of the beam is slightly bent. The optical properties of POF are, of course, related to the properties of the plastics themselves [19]. In our experiment, we used fibers from PMMA (polymethyl methacrylate), a synthetic polymer with thermoplastic properties which is used for the fiber core, with refractive indices around 1.49 and 1.59 [46]. The outer shell, however, is made of silicone resin with a refractive index of 1.46. It can be seen that a relatively large difference in refractive indices is introduced between the core and the outer shell [47]. The light transmittance of PMMA is about 92% in the entire range of the spectrum (this extends up to the UV region). At the same time, PMMA has good mechanical and electrical insulating properties, and is resistant to water, diluted alkalis and acids [44]. The most characteristic feature of PMMA is its clarity and complete lack of color even in thick layers. This allows not only for perfect transparency, but also for easy coloring. PMMA surpasses all conventional thermoplastics in its resistance to weathering [48]. Even after many years of exposure in a tropical climate, it was found to have only minimal changes in its clarity and color. POF also has a high numerical aperture. Plastics that have useful optical properties typically have a much lower density than optical glasses. POF is also immune to electrical interference and, at the same time, not electrically conductive [49]. POF made of PMMA with the technical parameters specified in the Table 3 and shown in the Figure 6 were used for our experimental measurements of light properties and the production of concrete slabs with optical fibers.

Since the monomode fibers propagate light in one clearly defined path, intermodal dispersion effects are not present, allowing the fibers to operate at larger bandwidths than a multimode fiber [50]. On the other hand, multimode fibers have large intermodal dispersion effects due to the many light modes of propagation they can handle at one time. The most interesting form of this phenomenon is probably the sharp display of shadows on the opposite side of the wall. Moreover, the color of the light also remains the same. A multimode fiber can propagate hundreds of light modes at one time while monomode fibers only propagate one mode as shown in the Figure 4 above [51].

For our experimental measurement of light properties and the production of concrete slabs with optical fibers we used POF type fibers made of PMMA. For a description of the specifications, see Table 3 below:

**Table 3 materials-16-03142-t003:** Plastic optical fibers (POF)—multimode—technical parameters.

POF—Multimode—Technical Parameters (Plastic Optical Fibers)	
Material:	Optical Loss (dB/km)
PMMA—polymethylmethacrylate	200
The greatest angle of incidence (°)	External refractive index:
75	n1 = 1.402 (glass n = 1.5–1.9)
Operating temperature (°C)	Refractive index of the core layer:
−10 + 70	n2 = 1.516
Range of wavelengths:	Minimum bending radius:
380–780	8 × D

**Figure 5 materials-16-03142-f005:**
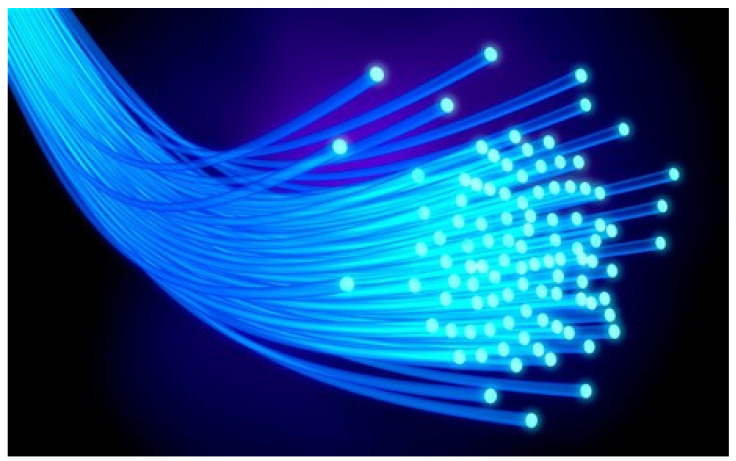
Optical fibers [52].

**Figure 6 materials-16-03142-f006:**
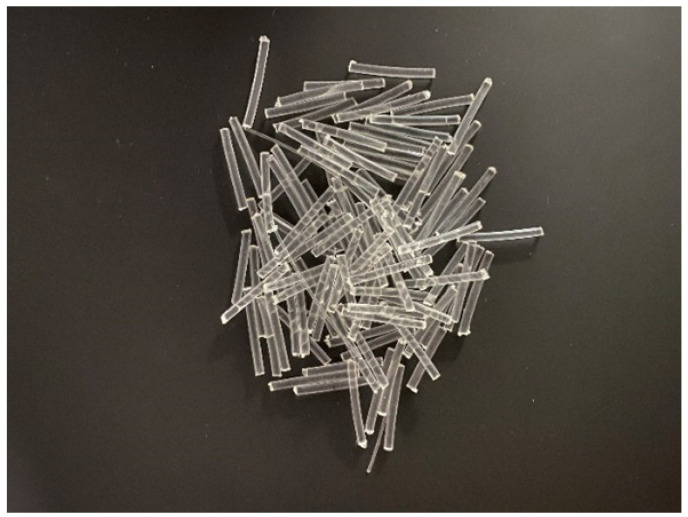
**Plastic** optical fibers (POF).

There are three basic types of optical fibers, as we can see in the Figure 7:(1)Multimode—Step-index fiber;(2)Multimode—Graded-index fiber;(3)Monomode—Single-mode fiber.

**Figure 7 materials-16-03142-f007:**
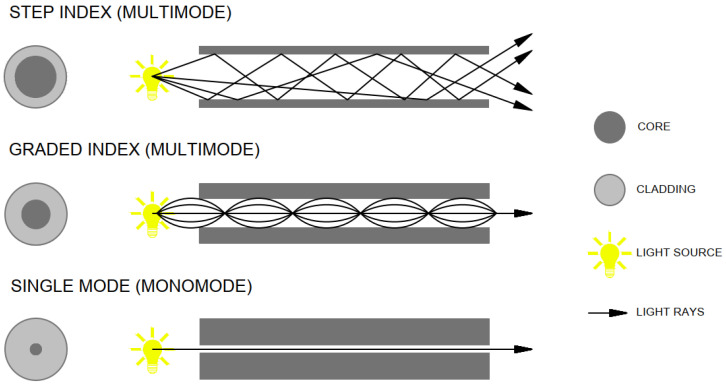
Types of optical fibers.

### 3.4. Design of the Samples

The shape of the sample that we constructed for our experiment was determined based on several factors: (1) the dimensions of the sample in relation to its particular use; (2) sample price; (3) sufficient strength; (4) transmission of light information and creation of a simple raster image. The dimensions were chosen to simulate standard construction practice. We tried to copy the dimensions of stone and ceramic cladding as much as possible. For that reason, we chose samples with dimensions of 250 × 250 mm with thicknesses of 20 mm to 30 mm for comparison. One of the main goals was to make samples for an acceptable price with sufficient strength and light-transmitting properties, which would then allow us to create a simple raster image. For the sake of comparison, we used the compressive and tensile flexural strength values of travertine stone. The definition of a raster image, or two-dimensional matrix of points (pixels), is when each point takes on certain values according to a simple image and forms a continuously filled area (raster). The bit depth defines the maximum number of shades. If we had wanted to achieve greyscale shades, we would have needed 8 bits. However, for our purposes, we wanted to achieve the color space, so we needed three RGB channels (red, green, blue), which requires 24 bits. To succeed, we had to place the optical fibers in as dense a grid as possible, while still optimizing the cost of production of the slabs. Another factor was the compaction of the concrete mixture with the glass fibers, so that it would spread evenly between the individual optical fibers and would not create air gaps that would weaken the strength of the slabs. Viewing distance is also very important. The further we are, the clearer the image will seem to us, and the closer we are, the more unclear it will appear. Thus, we chose an observation distance of 2 m from the light-transmitting slabs and adapted a grid of optical fibers of 3 mm thickness. The optimal distance of optical fibers in the sample grid—which is dependent upon the clarity of the image, the price of the slabs, and the feasibility of the construction—is 15 × 15 mm.

### 3.5. Samples Specification

Three types of samples of the same dimensions and two different widths were chosen for the production of unified slabs and subsequent experimental measurements. Individual slabs are shown in the Figure 8. All unified slabs were created from the mixture specified in article 2.2 using 3 mm diameter multimode POF made of PMMA, and Anti-Crak HP (62.4) glass fibers, which improve the mechanical properties of the boards, especially for carrying loads. All slabs were of unified dimensions, namely 250 × 250 mm with a thickness of 20 to 30 mm, specified in the Figure 9. The samples were divided into the following categories:

Category A:(1)Concrete slab with optical fibers of thickness 3 mm and Anti-Crak HP (62.4) glass fibers, dimensions of 250 × 250 × 20 mm, POF placed perpendicular to the exact grid at a distance of 15 × 15 mm, as shown in the Figure 10.(2)Concrete slab with optical fibers of thickness 3 mm and Anti-Crak HP (62.4) glass fibers, dimensions of 250 × 250 × 30 mm, POF placed perpendicular to the exact grid at a distance of 15 × 15 mm, as shown in the Figure 10.

Category B:(1)Concrete slab with air holes of thickness 3 mm and Anti-Crak HP (62.4) glass fibers, dimensions of 250 × 250 × 20 mm, air holes placed perpendicular to the exact grid at a distance of 15 × 15 mm, as shown in Figure 11.(2)Concrete slab with air holes of thickness 3 mm and Anti-Crak HP (62.4) glass fibers, dimensions of 250 × 250 × 30 mm, air holes placed perpendicular to the exact grid at a distance of 15 × 15 mm, as shown in the Figure 11.

Category C:(1)Concrete slab with Anti-Crak HP (62.4) glass fibers, dimensions 250 × 250 × 20 mm, as shown in the Figure 12.(2)Concrete slab with Anti-Crak HP (62.4) glass fibers, dimensions 250 × 250 × 30 mm, as shown in the Figure 12.

### 3.6. Production of Samples

A wooden form from medium density fiber board with a smooth surface and penetrations for optical fibers was constructed for the casting of translucent concrete, as in the Figure 13 and Figure 14. The plywood form was designed for the production of unified concrete slabs with dimensions of 250 × 250 × 20 mm and 250 × 250 × 30 mm, in which the optical fibers were laid in the transverse direction. Optical fibers were placed in a precise orthogonal grid and pulled through individual holes. The same procedure was repeated for the air-hole slabs, where the fibers were replaced by plastic tubes with steel wires, which were then removed. The mold was sufficiently sealed to prevent cement milk from leaking out of the mold during subsequent compaction. Next, we carefully poured the concrete mixture with glass fibers into the prepared forms between the optical fibers and the rubber tubes, which were reinforced with wires for their stability. We repeated this procedure gradually, pouring several layers. It was very important to compact the concrete mixture carefully, and to remove any air bubbles, which would have had a negative impact on compressive and tensile flexural strength. Compaction was complicated by the glass fibers, so great care needed to be taken to evenly distribute the mixture between the optical fibers and the rubber tubes with wires. Therefore, we used a vibrating table to compact the concrete mixture, which prevented any air pores from developing as the concrete was poured layer by layer. In addition, we added superplasticizers to better process the mixture and to spread it evenly throughout the mold. After the samples were removed from the framework, any protruding optical fibers were cut, and the slabs were sanded smooth to achieve similarity. Finally, the ends of the fibers were cleaned of dirt and any remnants of the concrete mixture and subsequently polished so that their permeability would be of the highest quality. One of the most important factors was a properly impregnated formwork for a sample of light-transmitting concrete, so, in order to create a product of greater quality than ordinary concretes, it was important that the surface was uniform, without air bubbles caused by poor vibration of the mixture in the formwork, so that the samples had the required surface quality shown in the Figure 15. Plastic optical fibers placed in the sample are shown in the Figure 16; rubber tubes with wires placed in the slab are shown in the Figure 17; and in the Figure 18 shows both elements used for making LTC samples: POF and rubber tubes with wires. We provided the specimens with conditions for ideal hardening of the mixture for a minimum of 48 h in a humid environment to acquire 50% of its strength, as shown in the Figure 19. Various created samples of the produced LTC unified slabs are shown in the Figure 20, Figure 21 and Figure 22.

## 4. Methods and Experimental Results

### 4.1. Experimental Part 1—Properties and Transmission

#### 4.1.1. Daylight Assessment

To assess the amount or quality of daylight, we worked with a calculation model. Our calculations considered the light scattered in the atmosphere, rather than the influence of direct sunlight, on the assessed spaces. Thus, we sought the least favorable sky condition, a uniformly cloudy sky in the winter with dark terrain. Because the sky is brighter toward the zenith, we considered the graded brightness of the sky. There are two graded brightness models which can be used. For dark terrain, CIE 1:3 is used, meaning that the zenith brightness is three times greater than the horizon brightness, while for the snowy terrain model, CIE 1:2 is used, which tells us that the zenith brightness is two times greater than the horizon brightness. The assessment of daylight is extremely difficult precisely because of these requirements, as the condition of uniformly overcast skies, according to the CIE, occurs only a few days a year. In order to determine a suitable lighting environment, quantitative and qualitative requirements must be fulfilled [53,54]. The daylighting of indoor spaces in buildings is designed and assessed according to several basic criteria, including: the level of daylight (expressed in terms of daylight factor values); the uniformity of illumination; glare; the distribution of luminous flux and the predominant direction of light; and the occurrence of other phenomena affecting visual comfort (e.g., the color of light and its reflection). In our experimental measurements, we assess the side illumination of a room through the perimeter wall of the building. In rooms with permanent human residents, which are illuminated by side light, at two control points at half the depth of the room, 1 m away from the inner surfaces of the side walls, the value of the daylight factor must be at least 0.7%, at the furthest 3 m away from the window, and the average value of these two points must be at least 0.9%. If the windows are in two adjacent walls, this requirement is sufficient for at least one of the two pairs of these control points [55]. When assessing the lighting parameters of the rooms, we followed the relevant standards that describe how to assess the lighting [56,57,58].

#### 4.1.2. Daylight Factor Assessment

To obtain the values of the daylight factor at the control points, the measurement of the daylight factor under the real sky was used. The daylight factor is not directly measured but is defined as a percentage expression of the illuminance at the control point *E* (lx) to the current horizontal exterior illuminance on an unshaded plane *E_h_* (lx). The instruments used for the measurements were a luxmeter and an illuminance meter. These devices will be discussed in the upcoming Section 4.1.6. In order for the measurement to have a meaningful value, we performed a measurement close to the desired condition, that is, in a uniformly cloudy sky in winter [53,54]. We measured under the real sky close to the conditions CIE 1:3. Due to the above-mentioned conditions of the cloudiness of the sky, the measurement was quite complicated. A scaled-down model of a reasonably sized room was used for the measurement, and a luxmeter sensor was subsequently placed in the model. It was important that all the materials used in the model corresponded to a real setting with their type of color and light reflection factors. To verify the measurement performed to obtain these factors, a control calculation method was used. There are some important factors to consider when evaluating light-transmitting structures. The light falling on the structure must therefore be divided into several parts: transmitted light, reflected light and absorbed light [59,60].

The spectral transmittance of the material *T* (*λ*) is defined as the ratio of the radiation *I_T_* (*λ*) passed through the sample to the intensity of radiation *I*_0_ (*λ*) incident on the sample, as shown in the Figure 23. The light-transmission performance of translucent concrete decreases as the angle of incidence of light increases; 30° is the limit of the acceptance angle of an optical fiber, and at this point most of the light is outside the acceptance cone, meaning that it contains the entire area of internal reflection [61].

#### 4.1.3. Quantitative and Qualitative Level of Daylight

The daylight factor *D* (%) is used to express the quantitative level of daylight. This factor is defined as a percentage expression of the illuminance at the control point *E* (lx) to the current horizontal exterior illuminance on an unshaded horizontal plane *E_h_* (lx). The measurement of the individual parameters must take place at the same time. The relationship from which the level of the daylight factor *D* (%) can be calculated is given in Equation (1) below:(1)$$D=\frac{E^{\;}}{E_{h}}\;\cdot 100\;(\%)$$

Illuminance *E_h_* (lx) is dependent on the average brightness of the sky *L_m_* (cd·m^2^*)* and is determined from the relationship in Equation (2) below:(2)$$E_{h}=\pi \cdot L_{m}\;(\mathrm{lx})$$

The quality criteria of daylighting include the distribution of light flux, the distribution of brightness of surfaces in the field of vision, prevention of glare, color design of the interior and uniformity of daylight.

The uniformity of daylight can be influenced by certain factors, such as the height of the room, bright paint, and the use of light-scattering materials like curtains and glass. The evenness of daylight is also influenced by the position of the windows (a higher location is preferable) and the color of the shading objects (the lighter the better) [32].

For our case, illuminance is one of the most important physical quantities to be detected. In practice, it is one of the most-used criteria in the evaluation of light.

The illuminance *E* is defined by the value of the luminous flux *dΦ* falling on an area of 1 m^2^. Illuminance is given in lux (lx), which represents a luminous flux of 1 lumen (lm) spread over an area of 1 m^2^. It can be calculated from the relationship in Equation (3) below, but in our case, it will be measured with the appropriate instruments.
(3)$$E=\frac{d\Phi }{dS}\;(\mathrm{lx})$$

#### 4.1.4. Light Reflection Factor Assessment

One of the key parts of this article is the measurement of the light reflection factor *ρ* (-) on the individual surfaces of the unified plates. This factor is based on the relationship in Equation (4) below:(4)$$\rho =\frac{\pi \cdot \;L}{E}\;(-)$$

Partial values for obtaining the light reflection factor were measured using an illuminance meter and a luxmeter. The measurements took place in such a way that the illuminance meter was aimed and focused in a direction perpendicular to the measured surface, and the surface brightness *L* (cd/m^2^) was taken. At the same time, the illuminance *E* (lx) of the given surface was measured using a luxmeter. During measurement, the sensor of the luxmeter could not interfere with the viewing angle of the illuminant, and at the same time the illuminant and the person operating the device could not shade the sensor of the luxmeter. Because the measurement of brightness and illuminance must be carried out simultaneously, it was necessary to engage the cooperation of two experimental workers. The values of the light reflection factor were normally proposed for the main surfaces of the interior spaces in average values, where we used the reflection factor value of 0.7 for ceilings, 0.5 for walls and 0.3 for floors or floor coverings.

#### 4.1.5. Measurement of the Light Transmission Factor

Another investigated parameter is the light transmission factor of filling the lighting holes, which was determined from the relationship mentioned in Equation (5) below:(5)$$\tau_{s,nor}=\frac{L_{s}}{L_{0}}\;(-)$$

First, the brightness of the sky or other background was measured through the illumination hole, in a perpendicular direction, using an illuminance meter. This gave the brightness value *L_s_* (cd/m^2^). After that, the aperture was opened as quickly as possible, and the brightness of the sky or other background was measured without the influence of the aperture, which gave us the brightness value *L*_0_ (cd/m^2^). Then the measured brightness values were inserted into relation *τ_S_*_,*nor*_
*= L_S_/L*_0_ to obtain the light transmission factor of the filling of the lighting opening *τ_s_*_,*nor*_. Measured values are shown in the Table 4.

#### 4.1.6. Measuring Devices and Aids

In order to obtain the input values of the light reflection factor of individual surfaces and the input data for determining the light transmission factor through the unified slabs using optical fibers, two measuring devices were used, namely an illuminance meter and a luxmeter. These measuring devices are the property of the Department of Building Structures of the Faculty of Civil Engineering of the Czech Technical University in Prague. Per the accuracy requirements, instruments were calibrated (corresponded to the accuracy) ČSN 36 0011-1 [62].

**Konica Minolta Luminance Meter LS—110**, which has the following parameters, was used to measure illuminance *L* (cd·m^2^):measuring angle 1/3°;viewing angle 9°;range 0.001 cd·m^2^ to 299.99 cd·m^2^;focusing distance from 1014 mm to infinity;relative spectral response of 8% of the CIE spectral luminous efficiency *V* (*λ*).

**Konica Minolta Illuminance Meter T—10 AM** was also used to measure illuminance. It is a luxmeter with a removable receptor head and has the following parameters:accuracy ±2% ±1 digital displayed value;equipped with a cosine deviation filter;range from 0.001 lx to 299.9 lx;relative spectral response of 6% of the CIE spectral luminous efficiency.

#### 4.1.7. Daylight Transmission Test of a Simulated Horizontal Structure Made of Light-Transmitting Concrete on a Down-Scaled Model of the Room

Weather conditions, which significantly affect the measurement itself, were documented prior at beginning. When we measured the daylight factor values, we chose a day with a uniformly cloudy sky, which forms a kind of diffuser that evenly scatters the light. At the same time, we tried to approximate the 5000 lx illuminance values on an outdoor unshaded horizontal plane when assessing illuminance, in addition to 20,000 lx of illuminance on an outdoor unshaded horizontal plane when assessing lighting quality, such as daylight uniformity. Measured values of the sky brightness are shown in the Table 5. Daily temperatures were 18 to 22 °C. The northeasterly wind was at a speed of 2 to 6 m/s. Relative air humidity was 83%. Each time, we first measured the brightness of the sky and then the illuminance at the control points. To simulate the passage of daylight through the light-transmitting concrete structure, we created a modified room model. The scaled-down model of a room of dimensions 250 × 250 × 250 mm was created at a scale of 1:20 and was therefore intended to symbolize a room of dimensions 5 × 5 × 5 m, with one of its perimeter walls made of light-permeable concrete; it is shown in the Figure 24. In our case, the ceiling structure was replaced by a concrete slab made of translucent concrete, through which light could pass into the interior. The scaled-down model was made of dark-colored wooden medium-density fiber boards to avoid light reflection as much as possible. The whole design of the model is shown in the Figure 25. During the measurement, the entire space was closed in order to limit the leakage of light or its penetration into the model. The ends of the optical fibers were carefully cleaned and polished so that the measurement could not be affected by contamination. Shading obstacles that could affect the measurement were eliminated in our case as we focused on the light values of the material itself.

For the horizontal construction, we used several types of boards, which we then measured separately through control points in a scaled-down model of the room. Subsequently, we placed the control points on the comparison plane of the room, or on the bottom plate of the reduced model of the room. Measurements were made using two luminometers, which simultaneously measured the illuminance at the point under consideration inside the room *E* (lx) and the exterior illuminance on the unshaded horizontal plane *E_h_* (lx). The layout of the measurement points is defined in the Figure 26. There were 7 control points. We used several types of slabs: (1) 20 and 30 mm thick concrete mix slabs with 3 mm diameter plastic optical fibers (POF); (2) 20 and 30 mm thick concrete mix slabs with 3 mm diameter air holes; and (3) solid 20 and 30 mm thick concrete mix slabs. Results of measurements of illumination are shown in the Table 6 and values of the reflection factor are shown in the Table 7.

The results of the daylight factor measurements are plotted and compared in the Figure 27. The results of the reflection factor measurements are plotted and compared in the Figure 28.

#### 4.1.8. Measurement of Light Transmission from an Artificial Light Source through a Simulated Vertical Structure from Light-Transmitting Concrete between Two Rooms in the Down-Scaled Model

Weather conditions, which significantly affect the measurement itself, were documented before the measurement. On the day of the measurement, there was a uniformly cloudy sky. Daily temperatures were 22 to 26 °C. The southeast wind was at a speed of 2 to 6 m/s. Relative air humidity was 91%.

Used light artificial source: RYET LED 400 lm; 4.4 W; 2700 K; 15,000 h; E14; 220–240 V~50/60 Hz; CRI > 80; 91 lm/W; LED1717G5.

For our measurement of light transmission from an artificial source, we used a similar scaled-down room model to measure the illumination as we can see in the Figure 29 and Figure 30. We researched light transmission through the dividing wall between two rooms without access to daylight, where the only light was from an artificial light source. Shading obstacles that could affect the measurement were eliminated as we focused on the light values of the material itself. A scaled-down model of two adjacent rooms with dimensions of 250 × 250 × 250 mm at a scale of 1:20 was chosen, and is shown in the Figure 31. We made the walls, floor and ceiling of the model from black boards. The rooms were 100% sealed against the access of any other light that could negatively affect the measurement. In general, for measurement, it was better to use a dark color, so that either there was either no light reflection or the values of reflection factor were close to zero. At the same time, the surface was not shiny, so that the measurement was as accurate as possible. During the measurement, the entire space was closed in order to limit the leakage of light or its penetration into the model, to assessed the amount of light that the source emitted and that passed through the construction simulating the wall. The ends of the optical fibers were carefully cleaned and polished so that the measurement could not be affected by contamination. We gradually moved the vertical structure away from the source by enlarging one room and changing the other. We gradually measured the values of light penetration into the other room and the light emitted in the room with the source, for both cases at individual control points. Thanks to the light-transmitting concrete construction, we were able to multiply the effectiveness of the artificial light source when the rays from this source illuminated not only the room with the light source, but also the adjacent room when passing through the light-transmitting concrete structure. We placed the control points on the comparison plane of the room, or on the bottom plate of the reduced model of the room. The layout of the measurement points is defined in the Figure 32. There were 7 control points. Due to the optical fibers, both natural and artificial light could transmit through the translucent concrete. The main goal was to reduce the lighting power consumption by using sunlight as a light source, or to use an artificial light source for more spaces, i.e., multiply its potential [63]. We used several types of slabs: (1) 20 and 30 mm thick concrete mix slabs with 3 mm diameter plastic optical fibers (POF); (2) 20 and 30 mm thick concrete mix slabs with 3 mm diameter air holes (3); and solid 20 and 30 mm thick concrete mixture slabs. The results of measurements of illumination are shown in the Table 8 and Table 9. Measured values of reflection factor are shown in the Table 10. The results of the illumination measurements are plotted and compared in the Figure 33. Measurements of the reflection factor are plotted and compared in the Figure 34.

### 4.2. Experimental Part 2—Strength Characteristics

#### 4.2.1. Characteristics of the Strength of Translucent Concrete Slabs

The samples used for our study were subjected to compressive and tensile flexural strength tests [64,65,66]. The results of these tests showed that the strength of the slabs made from the mixture containing the 3 mm thick optical fibers does not particularly reduce the strength of the samples themselves. The strengths of the created samples, both compressive strength and tensile flexural strength, can be compared with, for example, the strength of some types of stone, such as travertine stone, which is 76.1 MPa in compression strength, and 12.5 MPa in tensile flexural strength, or granite, which has a compression strength of 112.1 MPa, and a tensile flexural strength of 9.3 MPa. This claim is also supported by studies, which confirmed that translucent concrete does not lose its strength by adding optical fibers. The maximum 28-day compressive strength obtained was 48 MPa for TSC cubic specimens incorporating 4% volume of 3 mm diameter POF [67]. It was stated that translucent concrete samples (TCS) can provide a high light transmittance and good mechanical properties. TCS with a 4% POF volume ratio performs up to 21.4% of natural light transmittance and 24.7% of artificial light transmittance near the cube face, which is enough lighting for commercial and residential buildings [68]. It has also been confirmed in another study of TC slabs that compressive strength is not affected by adding optical or any other fibers. Mechanical properties decrease with increasing POF volume ratios and diameters, which in turn significantly increases light transmittance. It has been demonstrated that concrete mixtures can increase light transmittance to 21.4% with a volume ratio of 4% POF and 3 mm POF diameter. Good mechanical properties such as compressive strength are minimally affected [69].

#### 4.2.2. Measurement of Compressive Strength of Light Transmitting Concrete Slabs

The compressive strength of concrete is one of the initial material characteristics that is used in the classification of concrete and the calculation of other specific concrete properties. A compressive strength test diagram of light transmitting slabs is shown in the Figure 35. It is one of the basic mechanical properties that is determined by standard destructive tests on 150 × 150 × 150 mm cubes or cylinders with a diameter of 150 mm and a height of 300 mm. In our case, we assessed light-transmitting concrete slabs with dimensions of 250 × 250 × 20 mm or 250 × 250 × 30 mm. Before the actual test, the geometry of the test fixture must be verified. For a series of tests to determine the compressive strength of concrete slabs, 7 slabs with a thickness of 30 mm and 7 slabs with a thickness of 20 mm were produced. The slabs come in three types of design: slabs with glass and optical fibers, slabs with glass fibers and air holes and solid slabs with glass fibers. The compressive strength *f*_*c*,*k*_ is determined from the relationship in Equation (6) below:(6)$$f_{ck}=\frac{F}{A_{c}}=\frac{F}{b\cdot h}\;(\mathrm{MPa})$$
where *F* is the maximum load at specimen failure and *b*, *h* are the dimensions of the cross-sectional area of the test body to which the compressive load is applied.

The compressive strength of the LTC slabs was tested on samples using a Controls C68Z00 test press. The test samples were loaded sequentially until their failure in the compressive test. We removed all residual loose material and concrete pours from the surface of the sample, especially on the areas to be loaded and in contact with the press plates. The slabs were placed in the press at the center of the bottom pushed slab so that they were loaded in a direction perpendicular to the direction of concrete mixture placement. Next, we set a constant loading rate. We loaded continuously without any surges. Based on the measured values, we recorded the maximum load achieved. For light-transmitting concrete slabs, the stress increases uniformly up to the ultimate strength. Beyond the ultimate strength, destruction occurs by crushing or outright shattering of the material. Surface cracks usually appear but may not indicate major deformation or a breakdown in overall cohesion. The method of fracture is influenced by the structure of the material. The results of the compressive strength measurements are recorded in the Table 11 and Table 12, and evaluated in the Figure 36 and Figure 37. The process of the compressive strength testing of individual slabs is documented in the Figure 38.

#### 4.2.3. Measurement of Tensile Flexural Strength of Light Transmitting Concrete Slabs

The tensile flexural strength in bending is determined by a three-point bendonsample test. The three-point test arrangement is always a combination of bending and shear. The calculation of the tensile flexural strength is influenced by the assumption of stress distribution along the cross-section and the non-linear behavior of concrete. The tensile flexural strength of concrete is one of the basic mechanical properties that is determined on the samples. A principle of the test itself and the sample placement is shown in the Figure 39. In our case, we assessed light-transmitting concrete slabs with dimensions of 250 × 250 × 20 mm or 250 × 250 × 30 mm. Before the actual test, the geometry of the test fixture needed to be verified. For a series of three-point bending tests to determine the tensile flexural strength of concrete slabs, 5 slabs with a thickness of 30 mm and 8 slabs with a thickness of 20 mm were produced. The slabs came in three types of design: slabs with glass and optical fibers, slabs with glass fibers and air holes, and solid slabs with glass fibers. Plasticization of concrete and formation of microcracks occurs here. The tensile flexural strength of concrete for the three-point bending test is calculated from the relationship in Equation (7) below:(7)$$f_{cf}=\frac{3F\cdot \;l}{2b\cdot {\left(h\right)}^{2}}\;(\mathrm{MPa})$$
where *F* is the maximum load; *l*, *b* and *h* are the dimensions: span, width and height of the section.

The tensile flexural strength of the translucent slabs was tested on samples using a Galdabini Quasar 100measuring machine. The test samples were loaded sequentially until their failure in the tensile flexural test. We removed all residual loose material and concrete pours from the surface of the slabs, especially on the areas to be loaded by the 3-point bending test cylinders. The slabs were placed on the lower cylinders so that they were loaded in a direction perpendicular to the upper cylinder, as it shows on the 3-point bending test diagram. Next, we set a constant loading and a rate of 0.5 mm/min. We loaded continuously without any surges. Based on the measured values, we recorded the maximum load achieved. The measured resultant tensile flexural strengths from the 3-point bending test of individual light transmitting concrete slab are given in the in the Table 13 and Table 14, along with its properties and dimensions. The resulting strength is sufficient for use as a cladding material in the interior or as a cladding material on the façade. The strength is comparable to stone slabs commonly used in the construction industry.

Comparison of tensile flexural strengths of solid slabs with GF, slabs with POF and GF and slabs with air holes and GF is shown in the Figure 40 and Figure 41. As we can see in the Figure 42 and Figure 43, the tensile flexural strength of all slabs averages around 7.8 MPa for 30 mm slabs and 7.6 MPa for 20 mm slabs. Tensile flexural strength of solid slabs is on average 9.2 MPa for 20 mm thickness and 8.3 MPa for 30 mm thickness. On the contrary, the measured strength of slabs with optical fibers and glass fibers is around 6.8 MPa for 30 mm thickness and 8.8 MPa for 20 mm thickness. Finally, the strength of slabs with air holes and GF is in the range of 8.1 MPa for 20 mm thickness and 7.8 for 30 mm thickness.

## 5. Conclusions and Discussion

The aim of this paper was to determine the positive contributions made by light-transmitting concrete to the light values of building interiors, hence to the betterment of its overall environment. This in turn has a positive effect on the people who live and work in those buildings.

This paper presents an experimental study focused on the determination of daylight transmission of light through a horizontal structure and the transmission of light from an artificial source through a dividing structure between two spaces. In the experiment, test specimens were set up, on which the following light properties such as illumination, daylight factor and reflection factor were measured. Furthermore, the strength properties of LTC slabs in connection with their use as cladding, interior non-load bearing structures and facade.

We carried out measurements of the penetration of daylight through structures made of light-permeable concrete and the penetration of electromagnetic radiation from a stationary source through the dividing structure between two spaces. Based on the data from measurement, it can be said that the use of light-transmitting concrete does, in fact, support the supposition that interior light values do improve. It is well documented that the abundance of light benefits overall human emotional and mental health, and, for this important reason, we wanted to prove the usefulness of this material.

In communal spaces without access to daylight, the use of a light-transmitting concrete structure will significantly improve the lighting conditions. This is an improvement in units of percent, but in rooms where the level of daylight is at the minimum limit or in rooms without access to daylight, these are non-negligible values. It can be further determined that the thickness of the walls, partitions, ceiling, etc. does not have a negative effect on the transmission of light through the optical fibers placed in the light-permeable concrete structure. Regarding the comparison of light penetration through constructions with optical fibers compared to those with air penetrations, it was found that the daylight factor improved tenfold. The measured values of the daylight factor through the slabs with air holes are on average around 0.03%, in comparison with the values through the slabs with optical fibers, which are, on average, around 0.39%. So, if we build the perimeter walls of the rooms from light-permeable concrete, we obtain half the standard values of the daylight factor (*D*_min_ = 0.7% and *D*_aver_ = 0.9%), which is great for improving the interior space in relation to lighting. This could help indoor spaces in inner-city areas in terms of decreasing building setbacks. Structures made of light-transmitting concrete also have a secondary benefit, in the form of their aesthetic contribution to interior design.

The reflection factor of our samples of high-performance light-transmitting concrete with rough surfaces is around 0.5, and, for that with smooth surfaces, around 0.4. For comparison, the reflection factor of the black boards used to create the down-scaled model of rooms is around 0.01, which made them suitable to use for the model as they do not negatively affect the measured values.

In an experimental measurement of the light transmission from an artificial source from one room to another through a vertical partitioning structure, it was found that, on average, 5% of the light emitted from an artificial source passes through a structure made of light-permeable concrete. However, it depends upon the position of the artificial source relative to the optical fibers, the angle of the incident light to the ends of the optical fibers. When the angle of incident light is greater, less light passes through the structure. The light transmission decreases proportionally from the center of the circle created by the artificial light source, due to the light scattering on the slab surface. This effect, plus the angle of the incident light rays, results in minimal light transmission at the corners of the slab. The optical fibers that are placed perpendicular to the artificial light source transmit the highest amount of light through the slabs.

Regarding the placement of optical fibers in the concrete mixture, the measured results on the samples showed only a minimal deterioration of the compressive and tensile flexural strength. The experiment measured compressive strengths of the concrete, which were in the range of 57.2 MPa for the 250 × 250 × 20 mm high-performance light-transmitting concrete slabs, and around 65.9 MPa for the 250 × 250 × 30 mm slabs. For the use of the slabs as a cladding material on the interior structure or on the facade of the building, it was important to measure their tensile flexural strength. The values of tensile flexural strength for light-transmitting concrete slabs are in the range from 7.64 MPa for 20 mm thick slabs to 7.78 MPa for 30 mm thick slabs. By comparison, the compressive strength values of typically used cladding stone materials such as granite and travertine are 112.1 MPa and 76.1 MPa, respectively. The tensile flexural strength values are 12.5 MPa for travertine and 9.3 MPa for granite.

In conclusion, it is submitted that it would be wise to consider the use of light-transmitting concrete in construction to optimize its ability to contribute light from both the exterior, and between the interior spaces of buildings. Improving the amount of interior lighting, even if the increase is negligible compared to window openings, skylights, etc., has a positive effect on people, both psychologically and physically. Finally, we can say that the use of light-transmitting concrete structures, combining both optical and glass fibers together, will improve the lighting conditions of a building without significantly reducing the strength of the structure itself. This innovative element offers many possibilities for architectural use in the construction sector.

## Figures and Tables

**Figure 1 materials-16-03142-f001:**
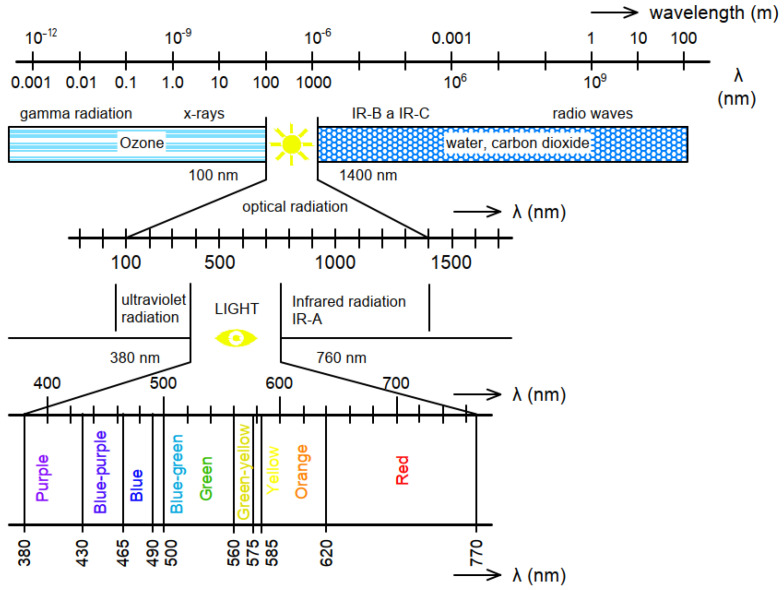
Electromagnetic spectrum.

**Figure 2 materials-16-03142-f002:**
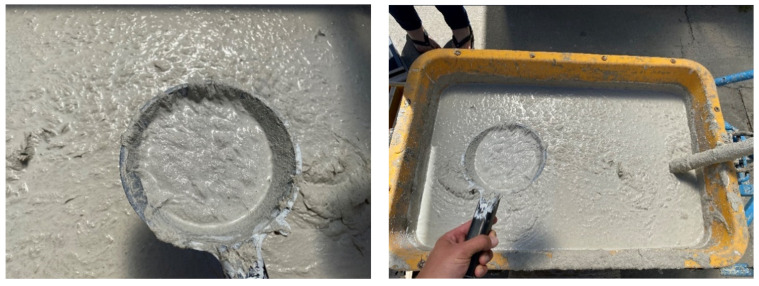
Concrete mixture for samples.

**Figure 3 materials-16-03142-f003:**
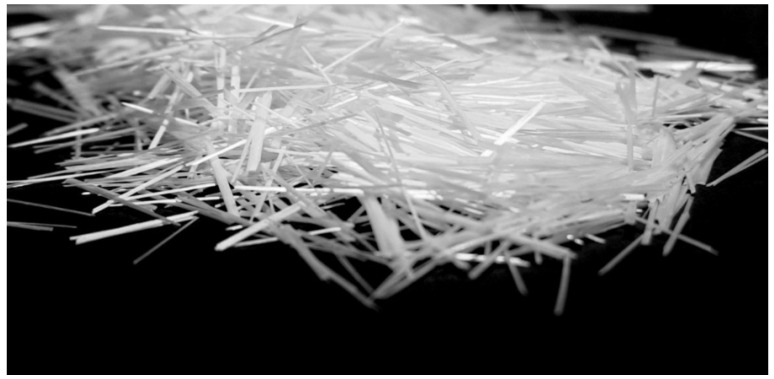
Glass fibers reinforcement.

**Figure 4 materials-16-03142-f004:**
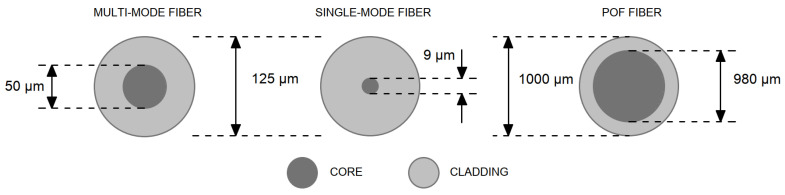
Comparison of glass and plastic fiber.

**Figure 8 materials-16-03142-f008:**
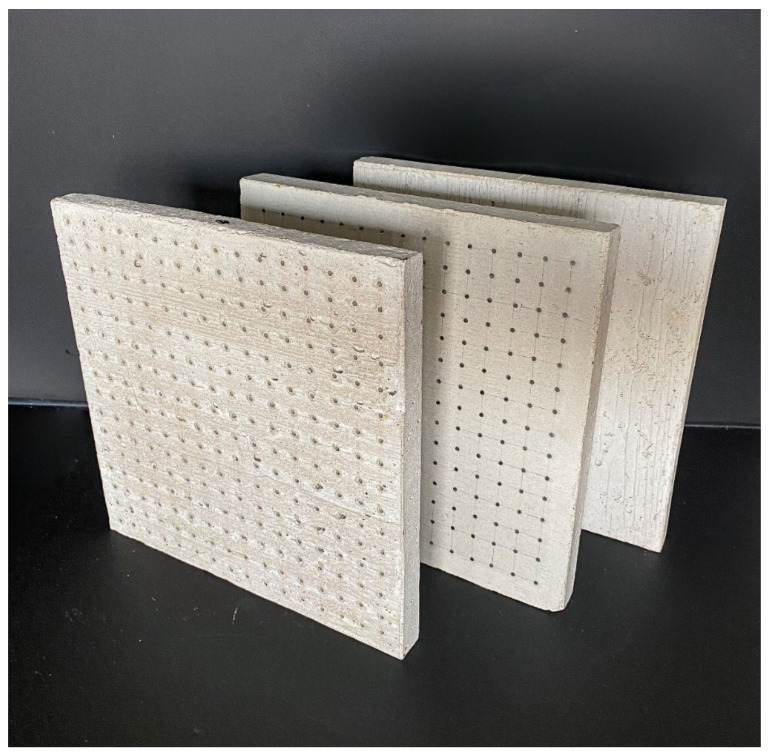
Samples of three types of used slabs.

**Figure 9 materials-16-03142-f009:**
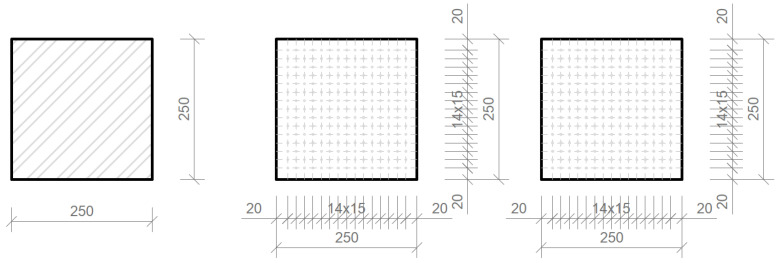
Drawings of individual slabs.

**Figure 10 materials-16-03142-f010:**
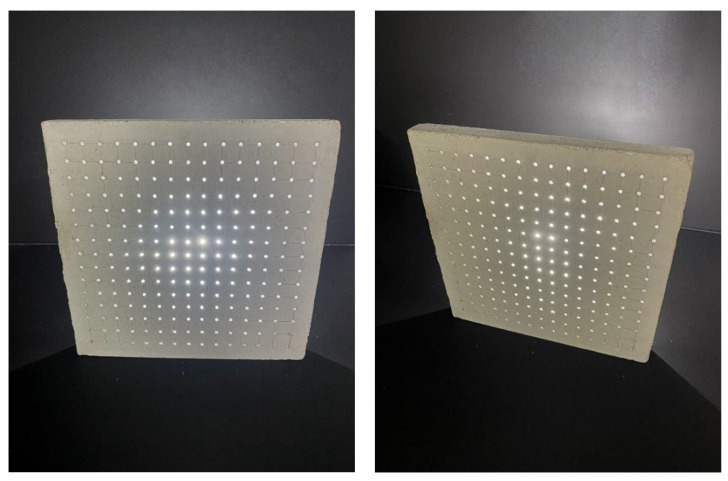
Concrete slab with optical fibers.

**Figure 11 materials-16-03142-f011:**
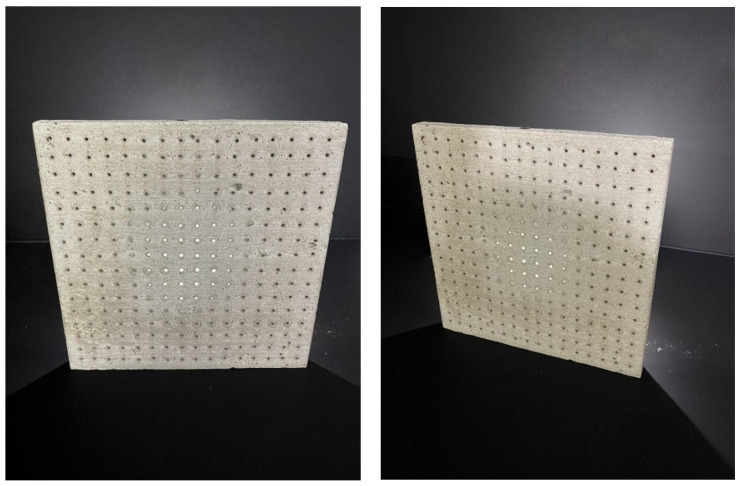
Concrete slab with air holes.

**Figure 12 materials-16-03142-f012:**
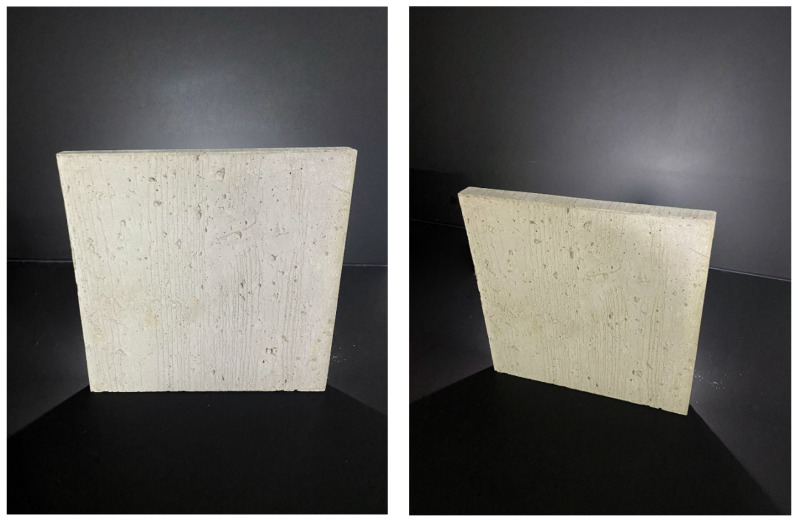
Concrete solid slab.

**Figure 13 materials-16-03142-f013:**
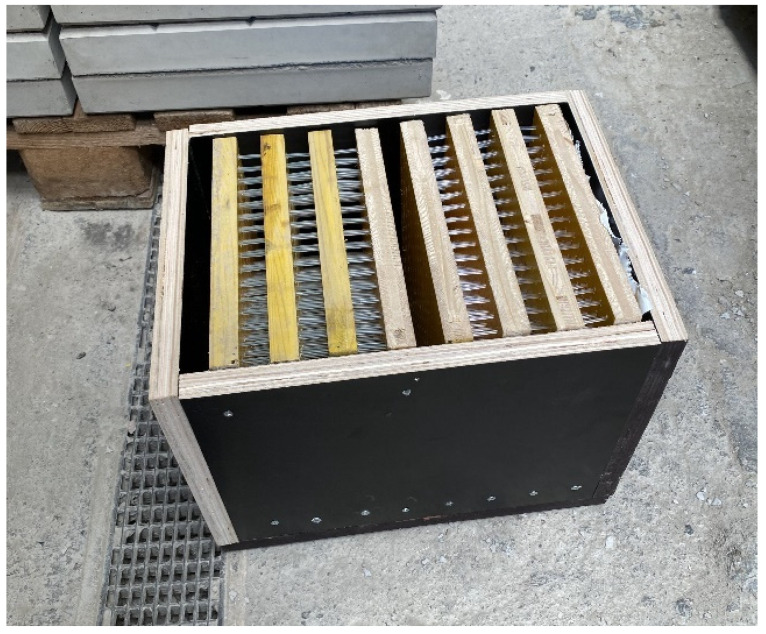
Formwork for making samples.

**Figure 14 materials-16-03142-f014:**
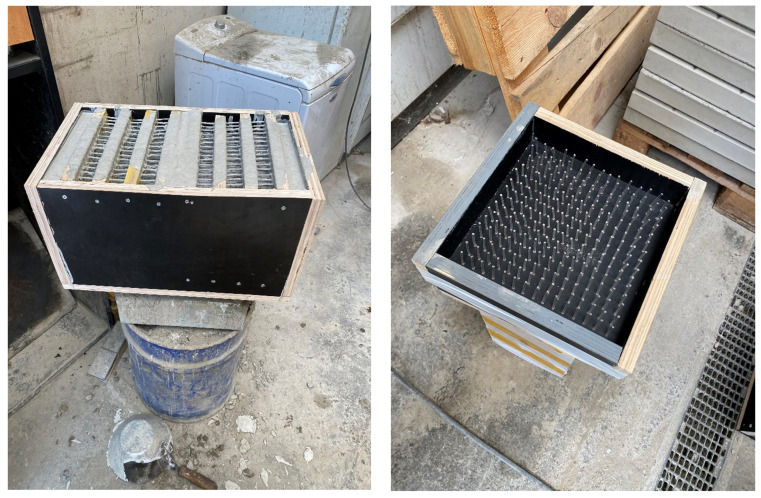
Formwork for making samples.

**Figure 15 materials-16-03142-f015:**
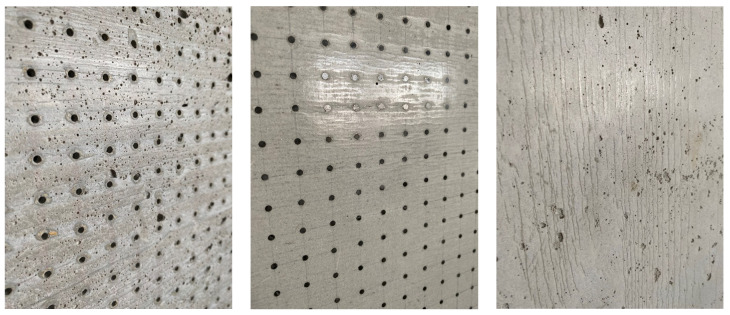
Examples of the surfaces of individual slabs of light-transmitting concrete.

**Figure 16 materials-16-03142-f016:**
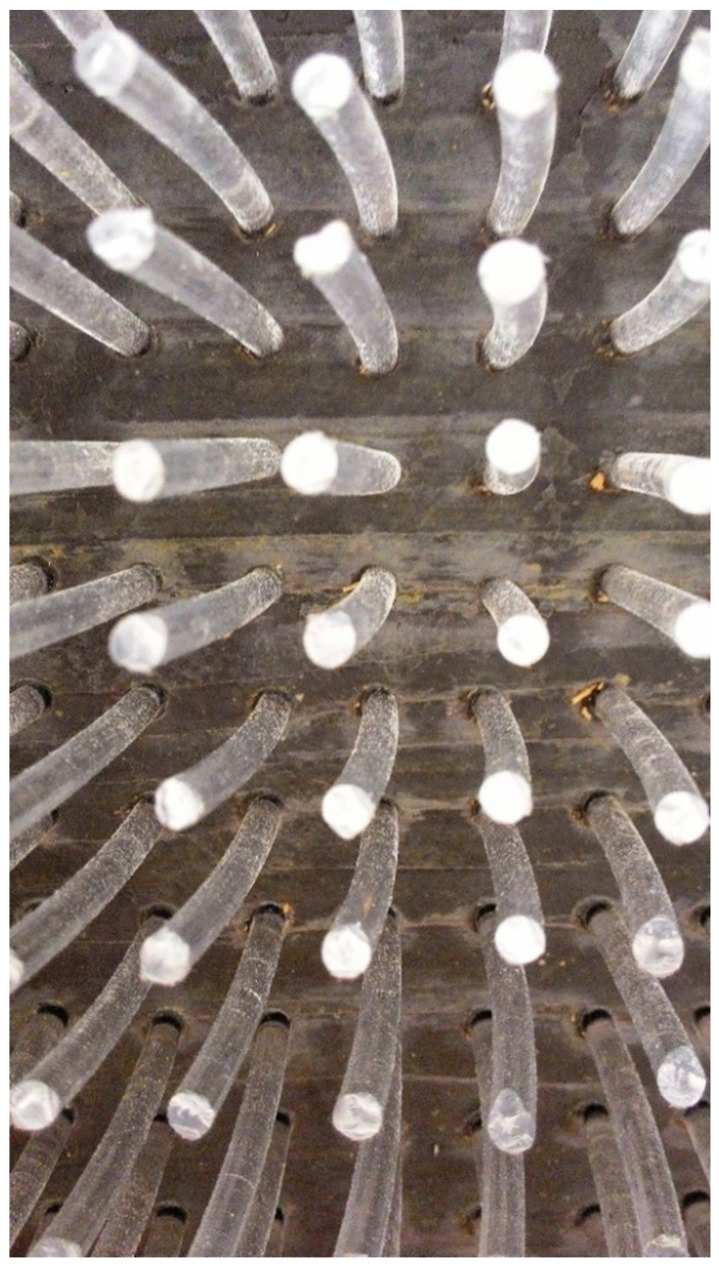
Sample with plastic optical fibers (POF).

**Figure 17 materials-16-03142-f017:**
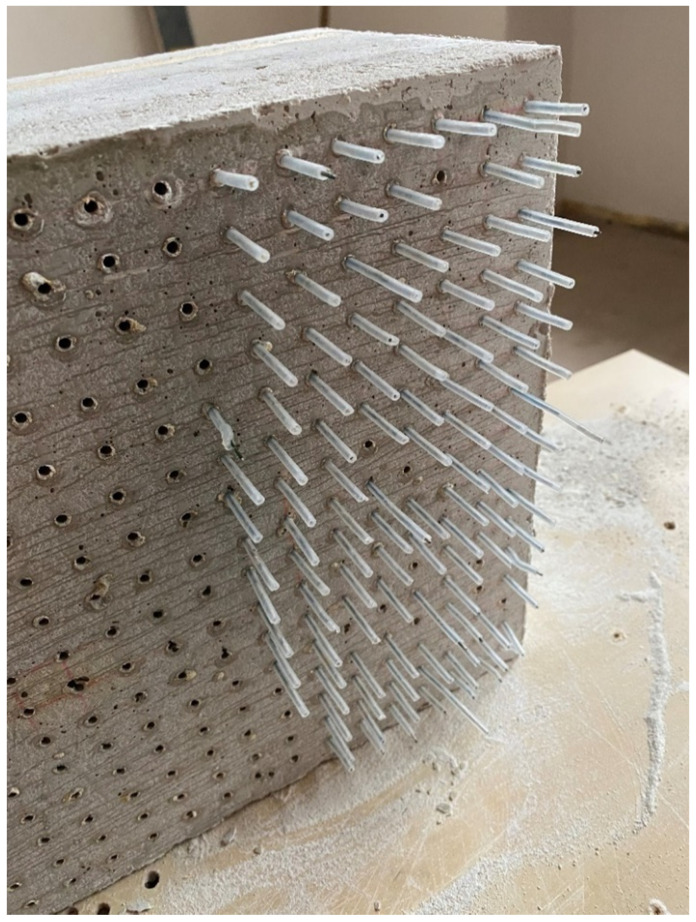
Sample with plastic tubes with wires.

**Figure 18 materials-16-03142-f018:**
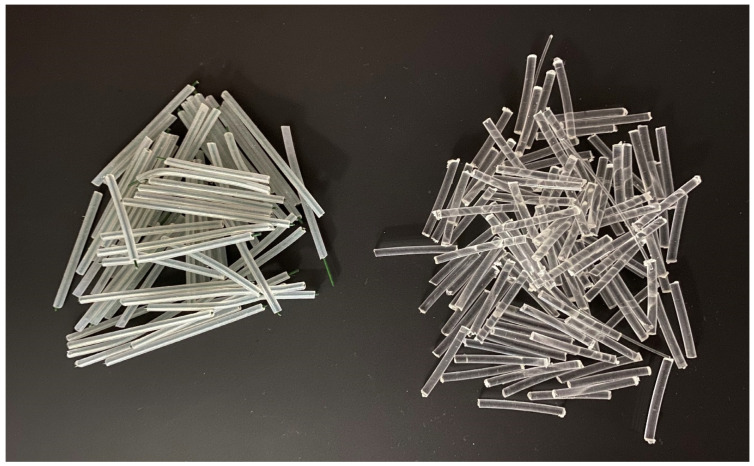
Plastic tubes with wires and plastic optical fibers (POF).

**Figure 19 materials-16-03142-f019:**
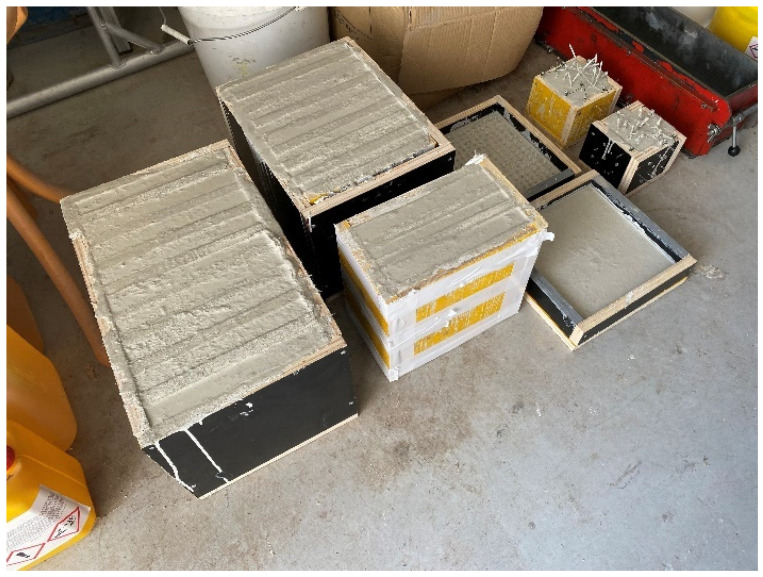
Formwork and hardening process of unified slab samples.

**Figure 20 materials-16-03142-f020:**
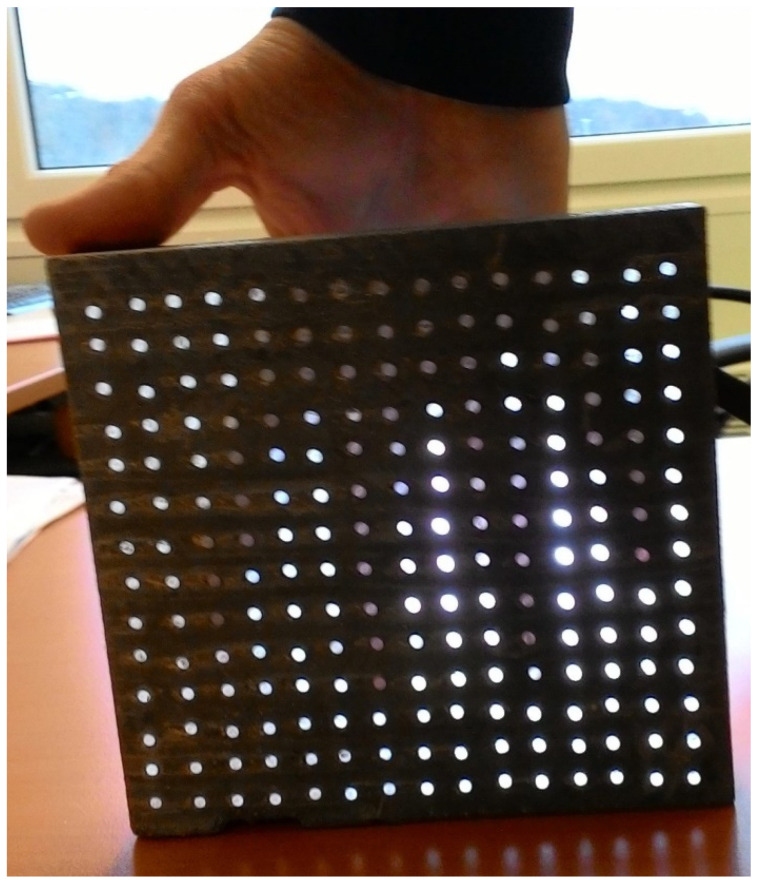
Example of light transmission through the slab.

**Figure 21 materials-16-03142-f021:**
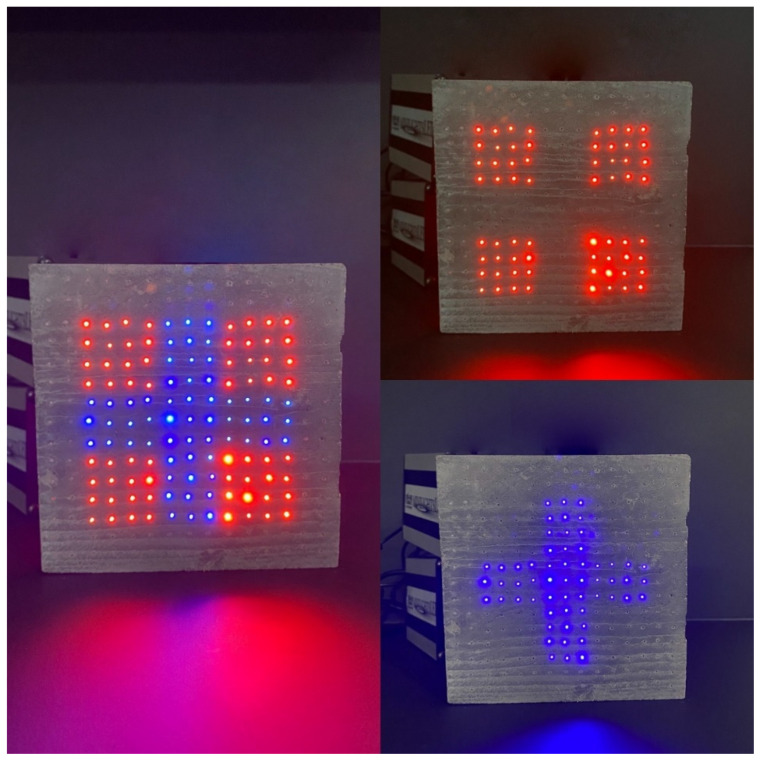
Examples of applications and lighting of light-permeable concrete slabs.

**Figure 22 materials-16-03142-f022:**
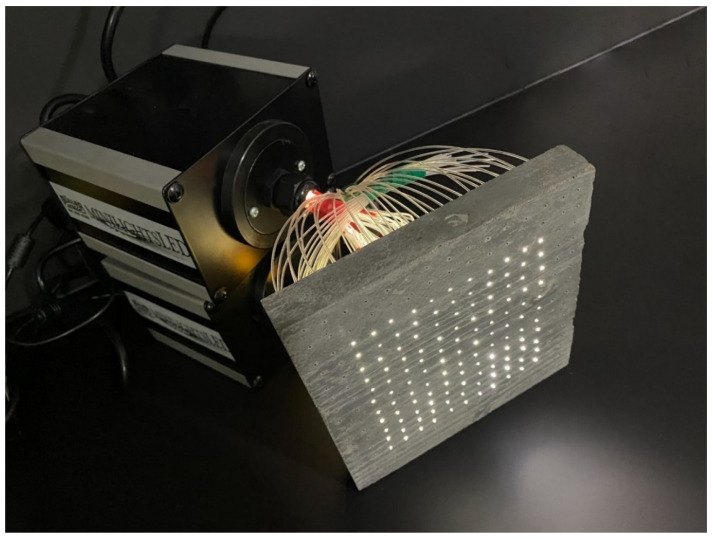
Slab with optical fibers connected to an artificial light source.

**Figure 23 materials-16-03142-f023:**
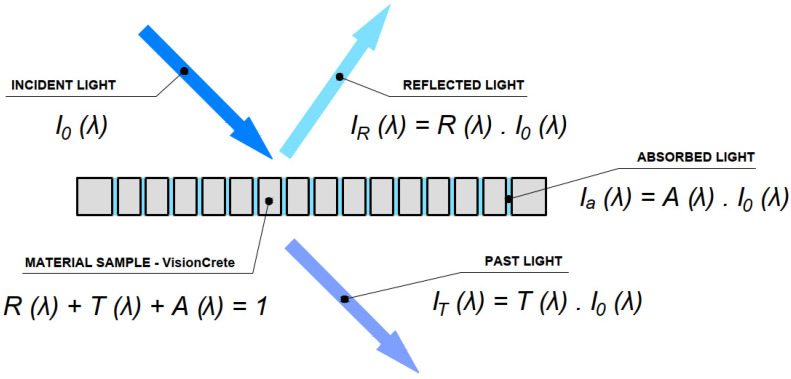
Diagram of the impact of the sun’s rays.

**Figure 24 materials-16-03142-f024:**
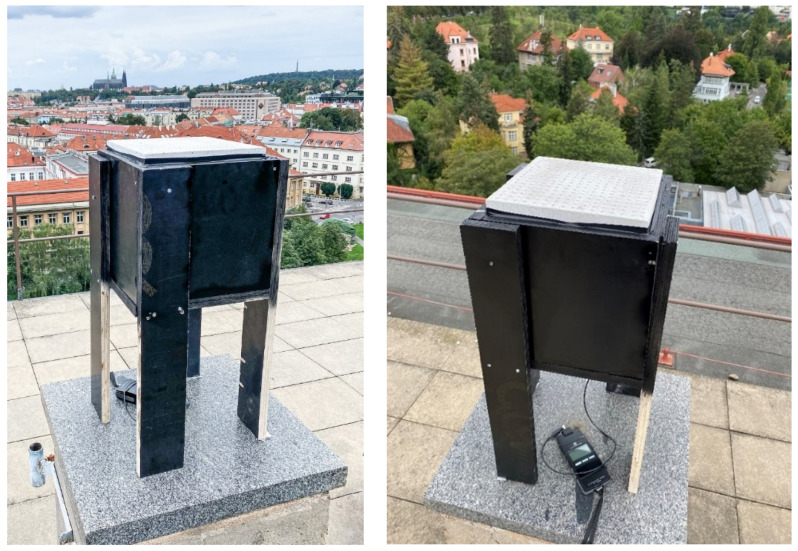
Down-scaled model for measurements.

**Figure 25 materials-16-03142-f025:**
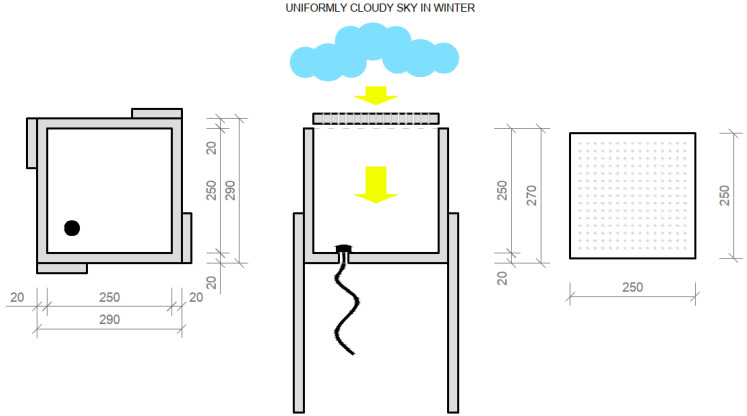
Plans and sections of the scaled-down model.

**Figure 26 materials-16-03142-f026:**
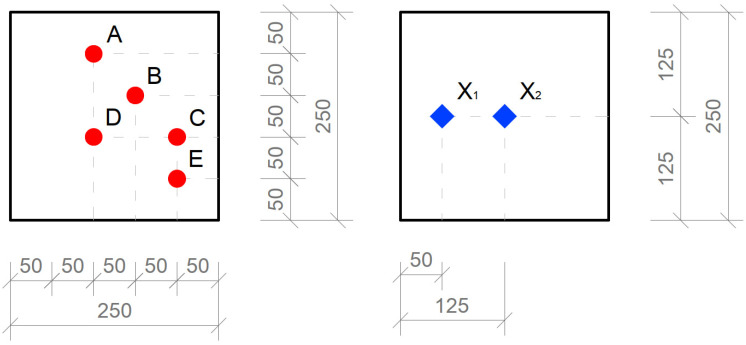
Drawing of measurement points of the horizontal plane of the model.

**Figure 27 materials-16-03142-f027:**
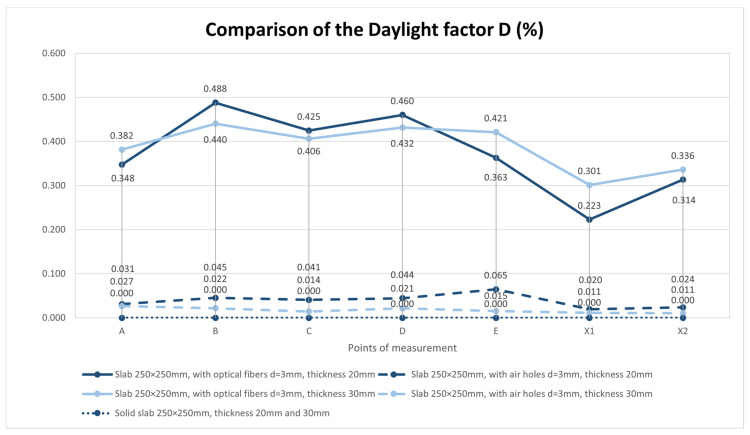
Comparison of the daylight factor *D* (%).

**Figure 28 materials-16-03142-f028:**
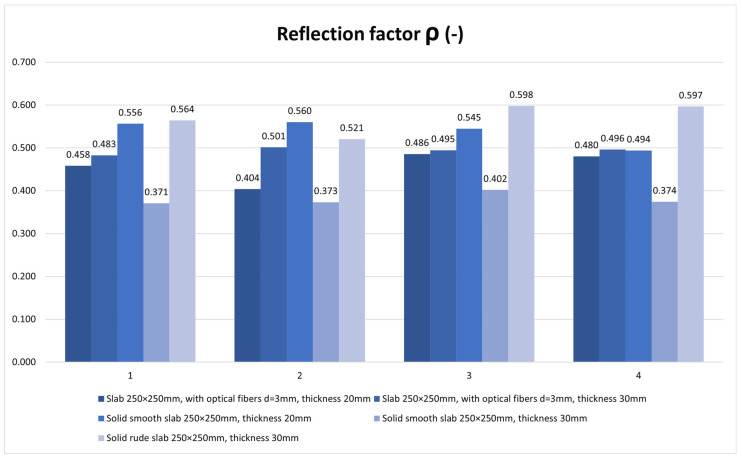
Reflection factor *ρ* (-).

**Figure 29 materials-16-03142-f029:**
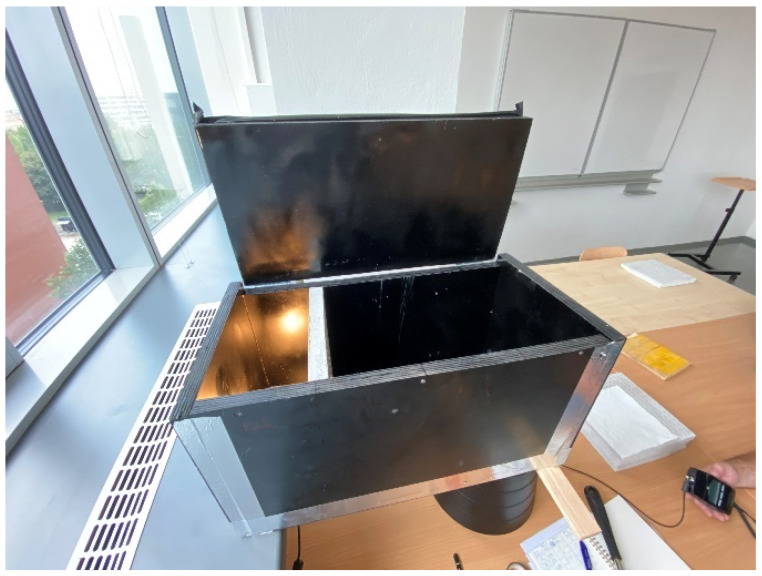
Down-scaled model for measurement.

**Figure 30 materials-16-03142-f030:**
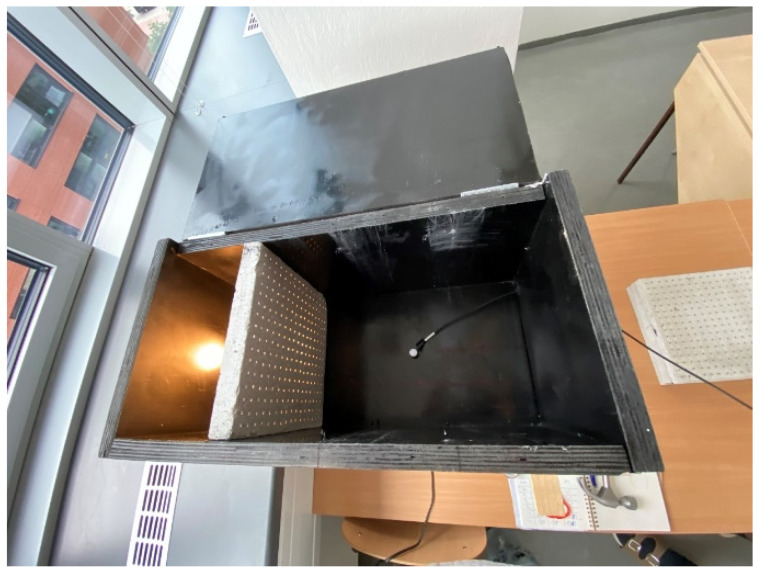
Down-scaled model for measurement.

**Figure 31 materials-16-03142-f031:**
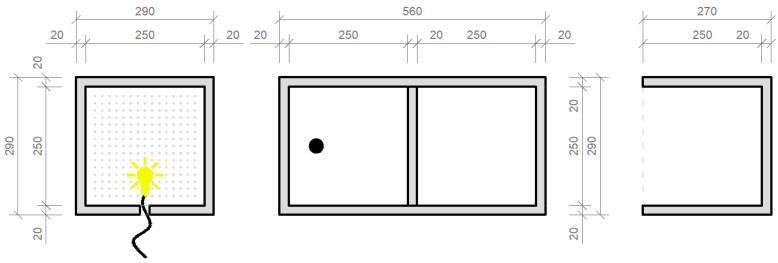
Plans and sections of the scaled-down model.

**Figure 32 materials-16-03142-f032:**
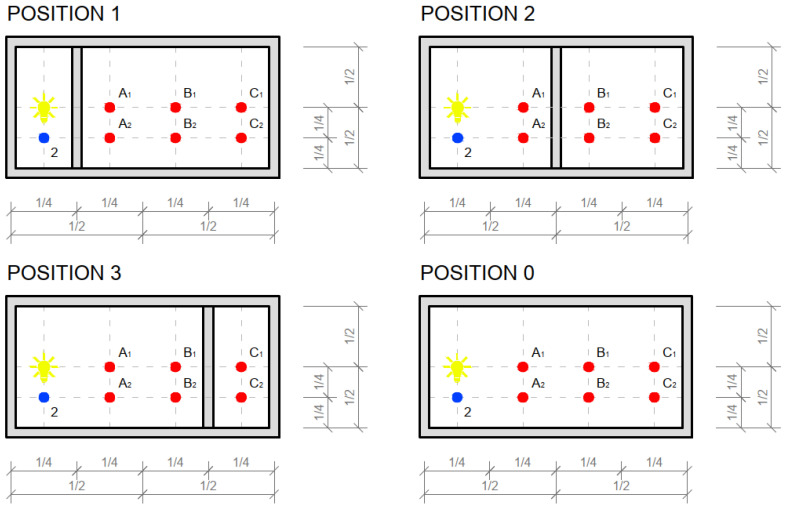
Drawing of measurement points.

**Figure 33 materials-16-03142-f033:**
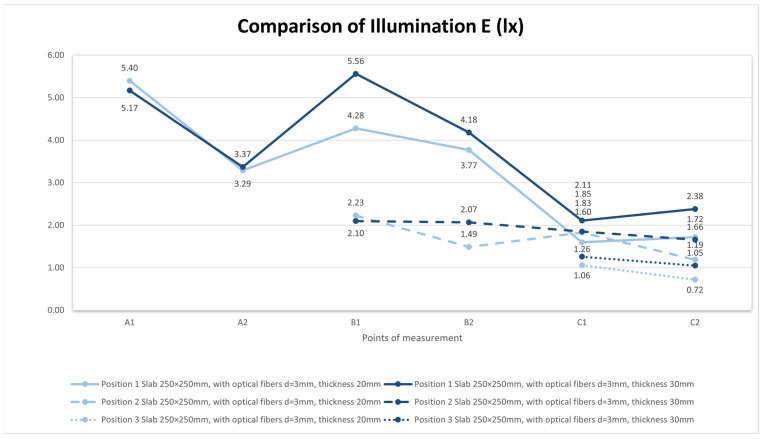
Illumination *E* (lx).

**Figure 34 materials-16-03142-f034:**
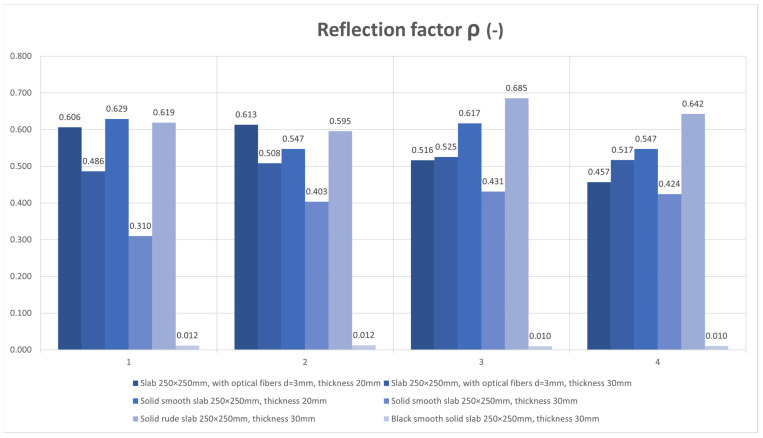
Reflection factor *ρ* (-).

**Figure 35 materials-16-03142-f035:**
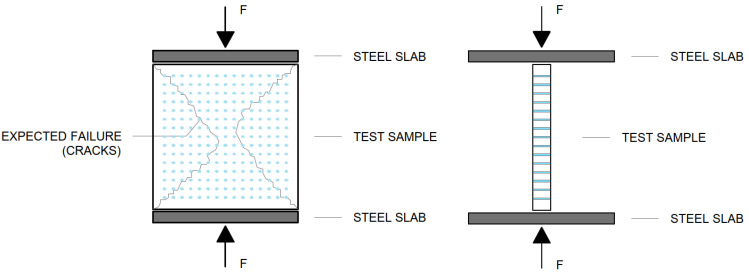
Compressive strength test diagram for LTC slabs.

**Figure 36 materials-16-03142-f036:**
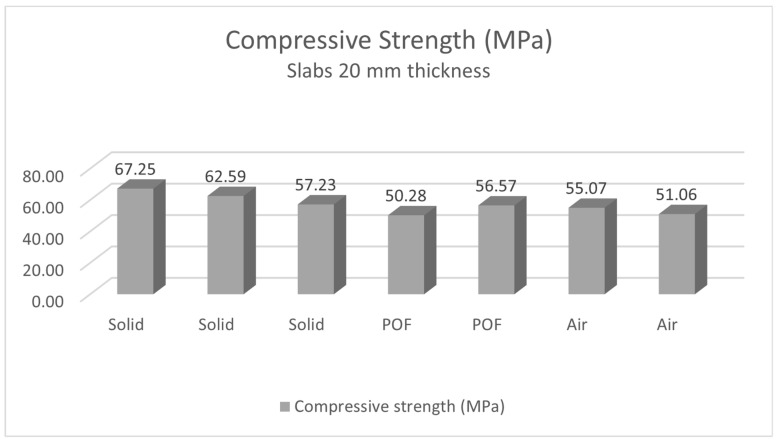
Compressive strength of slabs of thickness 20 mm.

**Figure 37 materials-16-03142-f037:**
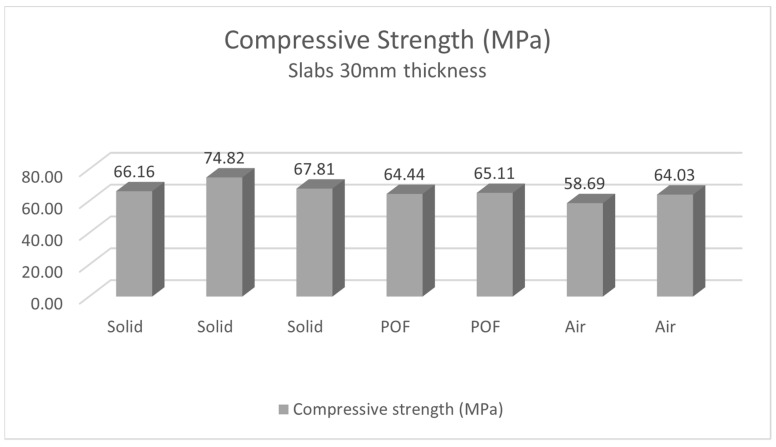
Compressive strength of slabs of thickness 30 mm.

**Figure 38 materials-16-03142-f038:**
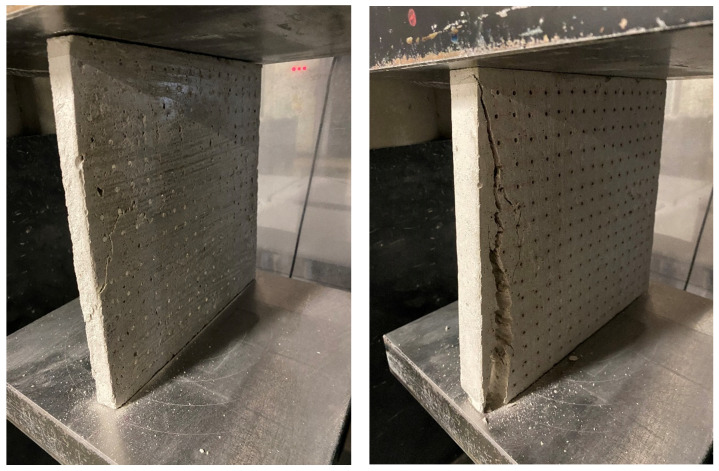
Compressive strength test—slab with POF and GF (**left**) and slab with air holes and GF (**right**).

**Figure 39 materials-16-03142-f039:**
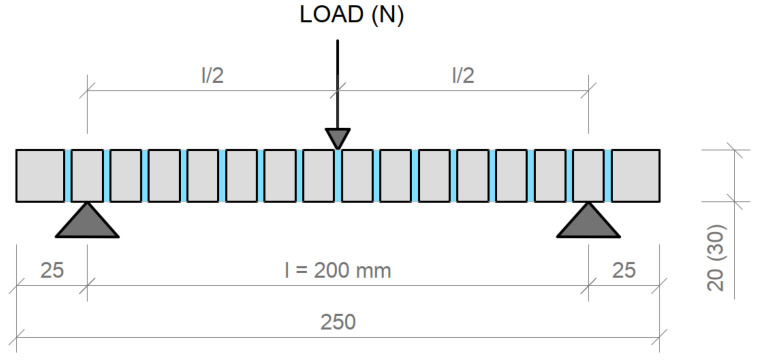
Tensile flexural strength 3-point bending test diagram for LTC slabs.

**Figure 40 materials-16-03142-f040:**
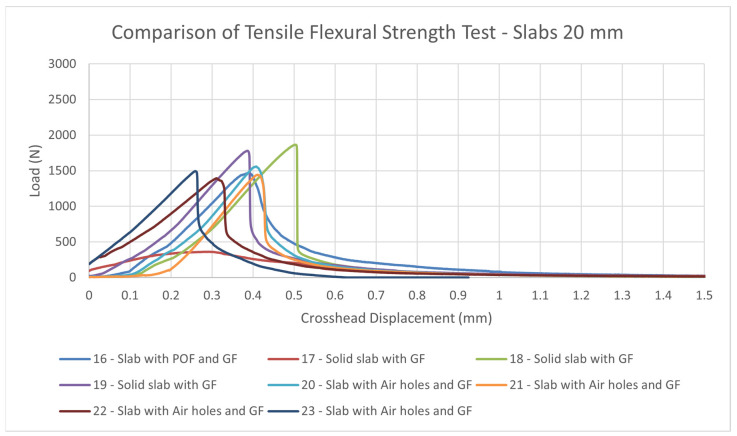
Comparison of tensile flexural strength test of slabs of thickness 20 mm.

**Figure 41 materials-16-03142-f041:**
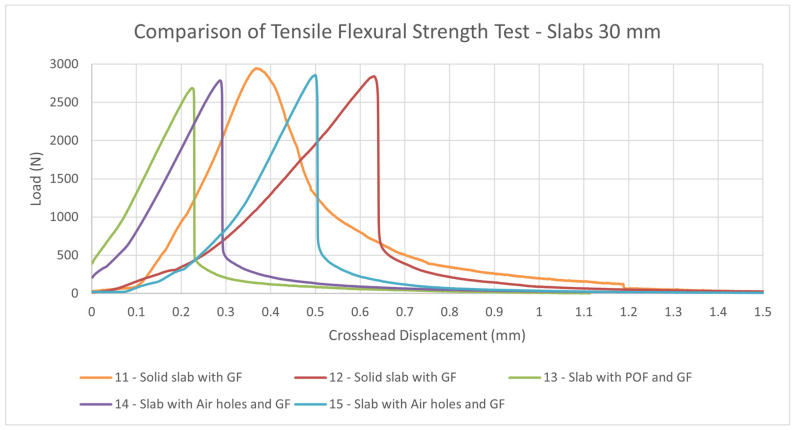
Comparison of tensile flexural strength test of slabs of thickness 30 mm.

**Figure 42 materials-16-03142-f042:**
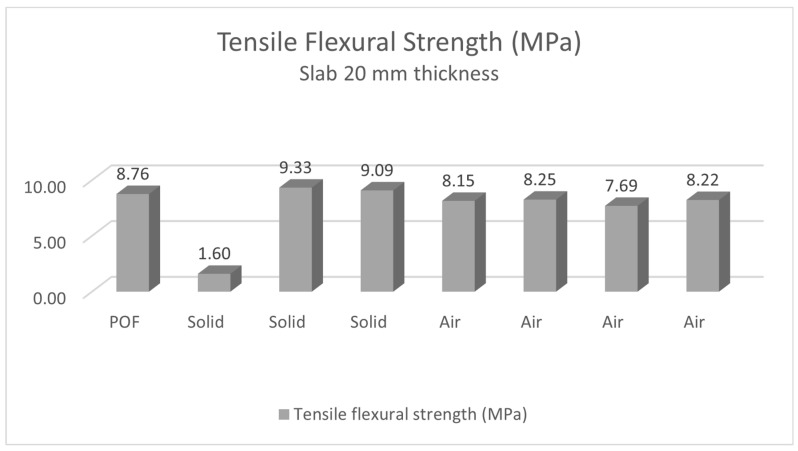
Tensile flexural strength of slabs of thickness 20 mm.

**Figure 43 materials-16-03142-f043:**
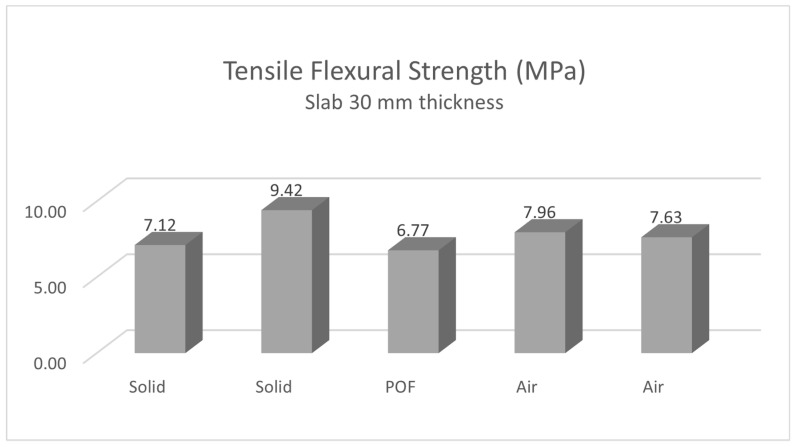
Tensile flexural strength of slabs of thickness 30 mm.

**Table 1 materials-16-03142-t001:** Components of the HPC mixture.

Components of the HPC Mixture	
Components	Approximate Density (kg/m³)
Aggregate 0–4 mm	947
White Portland Cement I 52.5 R	628
Bright microsilica	123
Superplasticizer	30
Glass fibers—Anticrack HP 12	20
Water	227
**TOTAL**	**1975**

**Table 2 materials-16-03142-t002:** Technical parameters of Anti-Crack HP (62.4) glass fibers.

Anti-Crak^®^ HP (62.4) Technical Parameters (Glass Fibers)				
Fiber Length (mm)	Filament Diameter (µm)	Tex	Lubricant Content (%) ISO 1887: 1980	Humidity (%) ISO 3344: 1977
6	14	45	0.08	0.3 max
12	14	45	0.08	0.3 max
Electrical conductivity: very low			Modulus of elasticity: 72 GPa	
Specific weight: 2.68 g/cm^3^			Spring tensile strength: 1700 MPa	
Material: alkali-resistant glass			Dosing of thin-walled prefabs up to 70 kg/m^3^	
Softening point: 860 °C				
Chemical resistance: very high				

**Table 4 materials-16-03142-t004:** Light transmission factor *τ*_*s*,*nor*_ (-).

Light Transmission Factor *τ*_*s*,*nor*_ (-):		
Values of the Light Transmission Factor when Measured against a Sunny Facade		
Slab 250 × 250 mm, with Optical Fibers d = 3 mm, thickness 20 mm		
*L_s_* (cd/m^2^)	*L_o_* (cd/m^2^)	*τ*_*s*,*nor*_ (-)
158.1	4733	0.033
181.7	1779	0.102
220.5	1398	0.158
Values of the light transmission factor when measured against a non-sunlit facade		
Slab 250 × 250 mm, with optical fibers d = 3 mm, thickness 20 mm		
*L_s_* (cd/m^2^)	*L_o_* (cd/m^2^)	*τ*_*s*,*nor*_ (-)
151.7	1166	0.130
387.5	1117	0.347
307.0	1075	0.286
Values of the light transmission factor when measured against a white wall in the room		
Slab 250 × 250 mm, with optical fibers d = 3 mm, thickness 20 mm		
*L_s_* (cd/m^2^)	*L_o_* (cd/m^2^)	*τ*_*s*,*nor*_ (-)
36.18	170.7	0.212
37.94	170.8	0.222
39.18	171.8	0.228

**Table 5 materials-16-03142-t005:** Distribution of the sky brightness.

Distribution of the Sky Brightness				
Zenith:	12,990	cd/m^2^	1874	cd/m^2^
Horizon:	3703	cd/m^2^	1180	cd/m^2^
	3892	cd/m^2^	998	cd/m^2^
	3357	cd/m^2^	1587	cd/m^2^
	3592	cd/m^2^	1624	cd/m^2^

**Table 6 materials-16-03142-t006:** Illumination *E* (lx) and daylight factor *D* (%).

Illumination *E* (lx) and Daylight Factor *D* (%):			
Slab 250 × 250 mm, with optical fibers d = 3 mm, thickness 20 mm			
Bod	*E* (lx)	*E_h_* (lx)	*D* (%)
A	18.30	5250	0.348
B	23.70	4860	0.488
C	22.00	5180	0.425
D	23.90	5190	0.460
E	21.60	5940	0.363
X_1_	10.70	4800	0.223
X_2_	15.10	4800	0.314
Slab 250 × 250 mm, with air holes d = 3 mm, thickness 20 mm			
Bod	*E* (lx)	*E_h_* (lx)	*D* (%)
A	1.80	5760	0.031
B	2.49	5500	0.045
C	2.40	5910	0.041
D	2.18	4920	0.044
E	3.78	5835	0.065
X_1_	0.90	4600	0.020
X_2_	1.14	4780	0.024
Slab 250 × 250 mm, with optical fibers d = 3 mm, thickness 30 mm			
Bod	*E* (lx)	*E_h_* (lx)	*D* (%)
A	18.70	4900	0.382
B	23.10	5250	0.440
C	23.10	5680	0.406
D	20.60	4760	0.432
E	22.20	5260	0.421
X_1_	14.00	4640	0.301
X_2_	16.00	4760	0.336
Slab 250 × 250 mm, with air holes d = 3mm, thickness 30 mm			
Bod	*E* (lx)	*E_h_* (lx)	*D* (%)
A	1.30	4880	0.027
B	1.14	5225	0.022
C	0.82	5710	0.014
D	0.96	4490	0.021
E	0.78	5110	0.015
X_1_	0.54	4720	0.011
X_2_	0.50	4740	0.011
Solid slab 250 × 250 mm, thickness 20 mm and 30 mm			
Bod	*E* (lx)	*E_h_* (lx)	*D* (%)
A	0.00	4790	0.000
B	0.00	5180	0.000
C	0.00	5670	0.000
D	0.00	4390	0.000
E	0.00	5080	0.000
X_1_	0.00	4670	0.000
X_2_	0.00	4690	0.000

**Table 7 materials-16-03142-t007:** Reflection factor *ρ* (-).

Reflection Factor *ρ* (-):		
Slab 250 × 250 mm, with optical fibers d = 3 mm, thickness 20 mm		
*E* (lx)	*L* (cd/m^2^)	*ρ* (-)
5010	730.6	0.458
5080	652.9	0.404
4960	767.0	0.486
4980	761.5	0.480
Slab 250 × 250 mm, with optical fibers d = 3 mm, thickness 30 mm		
*E* (lx)	*L* (cd/m^2^)	*ρ* (-)
4910	754.6	0.483
4930	786.6	0.501
4770	750.9	0.495
4690	740.7	0.496
Solid smooth slab 250 × 250 mm, thickness 20 mm		
*E* (lx)	*L* (cd/m^2^)	*ρ* (-)
4540	804.0	0.556
4450	793.6	0.560
4450	771.9	0.545
4400	691.7	0.494
Solid smooth slab 250 × 250 mm, thickness 30 mm		
*E* (lx)	*L* (cd/m^2^)	*ρ* (-)
4070	480.4	0.371
4030	478.9	0.373
3990	510.3	0.402
3920	467.0	0.374
Solid rude slab 250 × 250 mm, thickness 30 mm		
*E* (lx)	*L* (cd/m^2^)	*ρ* (-)
4410	792.1	0.564
4150	687.6	0.521
4110	782.0	0.598
4100	778.7	0.597

**Table 8 materials-16-03142-t008:** Illumination *E* (lx)—Part 1.

Illumination *E* (lx):			
Slab 250 × 250 mm, with optical fibers d = 3 mm, thickness 20 mm			
Point	Position 1	Position 2	Position 3
	*E* (lx)	*E* (lx)	*E* (lx)
A1	5.40	x	x
A2	3.29	x	x
B1	4.28	2.23	x
B2	3.77	1.49	x
C1	1.60	1.83	1.06
C2	1.72	1.19	0.72
Slab 250 × 250 mm, with air holes d = 3 mm, thickness 20 mm			
Point	Position 1	Position 2	Position 3
	*E* (lx)	*E* (lx)	*E* (lx)
A1	0.05	x	x
A2	0.02	x	x
B1	0.06	0.05	x
B2	0.04	0.03	x
C1	0.06	0.04	0.04
C2	0.05	0.04	0.04
Slab 250 × 250 mm, with optical fibers d = 3 mm, thickness 30 mm			
Point	Position 1	Position 2	Position 3
	*E* (lx)	*E* (lx)	*E* (lx)
A1	5.17	x	x
A2	3.37	x	x
B1	5.56	2.10	x
B2	4.18	2.07	x
C1	2.11	1.85	1.26
C2	2.38	1.66	1.05
Slab 250 × 250 mm, with air holes d = 3 mm, thickness 30 mm			
Point	Position 1	Position 2	Position 3
	*E* (lx)	*E* (lx)	*E* (lx)
A1	0.02	x	x
A2	0.00	x	x
B1	0.01	0.02	x
B2	0.00	0.00	x
C1	0.01	0.02	0.02
C2	0.00	0.01	0.02

**Table 9 materials-16-03142-t009:** Illumination *E* (lx)—Part 2.

Illumination *E* (lx):			
Empty test model/sample 500 × 250 × 250 mm		Empty test model/sample 250 × 250 × 250 mm	
Point	*E* (lx)	Point	*E* (lx)
A1	402.00	A1	442.00
A2	338.00	A2	368.00
B1	109.10	B1	x
B2	101.90	B2	x
C1	48.70	C1	x
C2	45.80	C2	x
1	position of light source	1	position of light source
2	698.00	2	730.00
Empty test model/sample 375 × 250 × 250 mm		Empty test model/sample 125 × 250 × 250 mm	
Point	*E* (lx)	Point	*E* (lx)
A1	394.00	A1	x
A2	342.00	A2	x
B1	107.10	B1	x
B2	103.30	B2	x
C1	X	C1	x
C2	X	C2	x
1	position of light source	1	position of light source
2	711.00	2	762.00

**Table 10 materials-16-03142-t010:** Reflection factor *ρ* (-).

Reflection Factor *ρ* (-):					
Slab 250 × 250 mm, with optical fibers d = 3mm, thickness 20 mm			Slab 250 × 250 mm, with optical fibers d = 3 mm, thickness 30 mm		
*E* (lx)	*L* (cd/m^2^)	*ρ*	*E* (lx)	*L* (cd/m^2^)	*ρ*
369	71.20	0.606	551	85.26	0.486
377	73.59	0.613	547	88.46	0.508
471	77.42	0.516	542	90.65	0.525
527	76.66	0.457	535	88.03	0.517
525	74.30	0.445	564	87.17	0.486
533	74.40	0.439	565	88.35	0.491
Solid smooth slab 250 × 250 mm, thickness 20 mm			Solid smooth slab 250 × 250 mm, thickness 30 mm		
*E* (lx)	*L* (cd/m^2^)	*ρ* (-)	*E* (lx)	*L* (cd/m^2^)	*ρ* (-)
454	90.90	0.629	444	43.79	0.310
477	83.09	0.547	406	52.13	0.403
450	88.38	0.617	425	58.32	0.431
456	79.39	0.547	418	56.45	0.424
Solid rude slab 250 × 250 mm, thickness 30 mm			Black smooth slab 250 × 250 mm, thickness 30 mm		
*E* (lx)	*L* (cd/m^2^)	*ρ* (-)	*E* (lx)	*L* (cd/m^2^)	*ρ* (-)
440	86.69	0.619	473	1.74	0.012
472	89.44	0.595	479	1.89	0.012
457	99.69	0.685	429	1.35	0.010
452	92.43	0.642	435	1.43	0.010

**Table 11 materials-16-03142-t011:** Properties of slabs 20 mm thick for measuring compressive strength.

Slabs 20 mm	Type of Slab	Length (mm)	Width (mm)	Thickness (mm)	Weight (kg)	Area (m^2^)	Load (kN)	Compressive Strength (MPa)
A	Solid	250	252	24.12	2.85	0.06	423.70	67.25
B	Solid	249	250	21.97	2.72	0.06	389.60	62.59
C	Solid	250	248	22.00	2.74	0.06	354.80	57.23
D	POF	254	250	19.30	2.29	0.06	319.30	50.28
E	POF	252	250	20.70	2.45	0.06	356.40	56.57
F	Air	249	250	20.62	2.51	0.06	342.80	55.07
G	Air	251	251	21.09	2.53	0.06	321.70	51.06

**Table 12 materials-16-03142-t012:** Properties of slabs 30 mm thick for measuring compressive strength.

Slabs 30 mm	Type of Slab	Length (mm)	Width (mm)	Thickness (mm)	Weight (kg)	Area (m^2^)	Load (kN)	Compressive Strength (MPa)
1	Solid	251	252	28.90	3.49	0.06	418.50	66.16
2	Solid	251	251	36.35	4.13	0.06	471.40	74.82
3	Solid	252	249	33.80	4.09	0.06	425.50	67.81
4	POF	250	253	29.79	3.67	0.06	407.60	64.44
5	POF	250	253	29.51	3.69	0.06	411.80	65.11
6	Air	250	253	29.95	3.69	0.06	371.20	58.69
7	Air	250	252	29.71	3.65	0.06	403.40	64.03

**Table 13 materials-16-03142-t013:** Properties of slabs 20 mm thick for measuring tensile flexural strength.

Slabs 20 mm	Type of Slab	Length (mm)	Width (mm)	Thickness (mm)	Weight (kg)	Area (m^2^)	Load (kN)	Tensile Flexural Strength (MPa)
16	POF	252	121	20.38	1.20	0.03	1467.30	8.76
17	Solid	252	124	23.41	1.47	0.03	362.20	1.60
18	Solid	250	125	21.90	1.36	0.03	1865.03	9.33
19	Solid	257	123	21.86	1.32	0.03	1780.27	9.09
20	Air	250	125	21.43	1.27	0.03	1558.75	8.15
21	Air	251	122	20.73	1.21	0.03	1442.51	8.25
22	Air	251	124	20.93	1.24	0.03	1393.22	7.69
23	Air	250	124	20.97	1.26	0.03	1494.15	8.22

**Table 14 materials-16-03142-t014:** Properties of slabs 30 mm thick for measuring tensile flexural strength.

Slabs 30 mm	Type of Slab	Length (mm)	Width (mm)	Thickness (mm)	Weight (kg)	Area (m^2^)	Load (kN)	Tensile Flexural Strength (MPa)
11	Solid	250	122	31.89	2.10	0.03	2944.63	7.12
12	Solid	249	126	26.81	1.71	0.03	2842.74	9.42
13	POF	251	126	30.72	1.89	0.03	2685.12	6.77
14	Air	251	125	28.97	1.81	0.03	2784.94	7.96
15	Air	251	122	30.34	1.79	0.03	2856.64	7.63

## Data Availability

Data will be made available upon reasonable request.

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
