# Peer review of "Illumination of Interior Spaces through Structures Made of Unified Slabs of High-Performance Light-Transmitting Concrete with Embedded Optical Fibers"

_materials, 2023, doi:10.3390/ma16083142_

Round 1

Reviewer 1 Report

The paper presented an experimental test on light transmitting concrete by a down-scaling method. The study is of importance; however, the current form of the paper is not acceptable. Too many formatting errors, lack of discussions on the important section and figures and tables lacks description. Even worse, figures and tables are not mentioned in the text.

Technical aspects need further clarification, one major concern is the concrete density where it is less than 2400 kg/m3 (to be exact 1892 kg/m3) and therefore, is not categorised as normal weight concrete.

Other detailed comments are given in the attachment. In the current forms, the paper needs a total revamp before it can be accepted for publication.

Author Response

Dear Reviewer,

I would like to thank you for your substantive comments on my paper, which I have tried to correct and incorporate in the text. I really appreciate your commentary, which I hope will lead to the improvement of the paper.

My responses to each of your comments are in the attachement.

I would like to thank you for your cooperation.

Reviewer 2 Report

I recommend the paper " Illumination of interior spaces through the structures made of unified slabs from high-performance light transmitting concrete with embedded optical fibers” for publication. Nevertheless, the manuscript required minor revision.

 The manuscript focused on the illumination of interior spaces using constructions made of light-transmitting concrete, which will allow light to pass between individual spaces. The Authors measure the improvement of lighting conditions inside a building using light-transmitting concrete structures. The paper potentially contributes to the literature as it presents results of interest for practice engineering and scientific purposes. Nevertheless, the manuscript required minor revision.

 General report and comments:

Introduction. Please extend a literature review in the investigated field.

Chapter 4. Please numbered all equations.

Conclusion chapter. Please, summarize the conclusions using bullet points. It would certainly emphasize the significance of the outcomes. Additionally, there should be closing remarks after the general conclusions (after the bullet points of conclusions), keeping in mind all the outcomes obtained. Please rebuild and supplements the conclusions. I hope the authors can try to get more scientific conclusions from the performed investigations.

Author Response

(The authors gave the same response as above.)

Round 2

Reviewer 1 Report

The authors had made the necessary amendments based on the previous comments by the Reviewer. However, the Reviewer found that the reference lists should be written in alphabetical sequence order from A-Z. Therefore, the paper can only be accepted once the formatting had been thoroughly checked by the Auhtors.

Author Response

Dear Reviewer,

I would like to thank you for your comment on my paper especially about reference list.

The authors had made the necessary amendments based on the previous comments by the Reviewer. However, the Reviewer found that the reference lists should be written in alphabetical sequence order from A-Z. Therefore, the paper can only be accepted once the formatting had been thoroughly checked by the Authors.

I have tried to edited references in article. I followed the guide for the authors MDPI journal, about references. (https://www.mdpi.com/journal/materials/instructions#references), but if I am to follow the citation style of the journal, it is not possible to arrange the citations alphabetically.

References must be numbered in order of appearance in the text (including table captions and figure legends) and listed individually at the end of the manuscript. In the text, reference numbers should be placed in square brackets [ ], and placed before the punctuation; for example [1], [1–3] or [1,3].

I have chosen numerical numbering of citations in the text as they appear sequentially, so it is not possible to keep the alphabetical order of citations. Thank you for your understanding.

I would like to thank you for your cooperation.
